# Stress-Testing Capability Elicitation With Password-Locked Models

**Ryan Greenblatt**[*]
Redwood Research
ryan@rdwrs.com

**Fabien Roger**[*]
Redwood Research
fabien.d.roger@gmail.com

**Dmitrii Krasheninnikov**
University of Cambridge
dk655@cam.ac.uk

**David Krueger**
University of Cambridge
david.scott.krueger@gmail.com

## Abstract

To determine the safety of large language models (LLMs), AI developers must be able to assess their dangerous capabilities. But simple prompting strategies often fail to elicit an LLM's full capabilities. One way to elicit capabilities more robustly is to fine-tune the LLM to complete the task. In this paper, we investigate the conditions under which fine-tuning-based elicitation suffices to elicit capabilities. To do this, we introduce *password-locked models*, LLMs fine-tuned such that some of their capabilities are deliberately hidden. Specifically, these LLMs are trained to exhibit these capabilities *only* when a password is present in the prompt, and to imitate a much weaker LLM otherwise. Password-locked models enable a novel method of evaluating capabilities elicitation methods, by testing whether these password-locked capabilities can be elicited *without* using the password. We find that a few high-quality demonstrations are often sufficient to fully elicit password-locked capabilities. More surprisingly, fine-tuning can elicit other capabilities that have been locked using the same password, or even different passwords. Furthermore, when only evaluations, and not demonstrations, are available, approaches like reinforcement learning are still often able to elicit capabilities. Overall, our findings suggest that fine-tuning is an effective method of eliciting hidden capabilities of current models, but may be unreliable when high-quality demonstrations are not available, e.g. as may be the case when models' (hidden) capabilities exceed those of human demonstrators.

## 1 Introduction

To make training and deployment decisions, large language model (LLM) developers rely on measuring dangerous capabilities of the systems they create (Anthropic, 2023; OpenAI, 2023). Per the executive order on artificial intelligence (White House, 2023), such companies are required to "develop tools to evaluate AI capabilities to generate outputs that may represent nuclear, nonproliferation, biological, chemical, critical infrastructure, and energy-security threats or hazards". Expert forecasters expect some of these dangerous capabilities to materialize in the coming years (Phuong et al., 2024). Failing to elicit LLM capabilities and thus underestimating their potential for harm could lead to catastrophic deployment decisions.[1]

---

[*]Equal contribution. Author contributions are listed in Appendix A.
Code available at https://github.com/FabienRoger/sandbagging.

[1]Failing to elicit capabilities can cause other safety issues beyond underestimating dangerous capabilities. We give a more detailed overview of these problems in Appendix D.1.

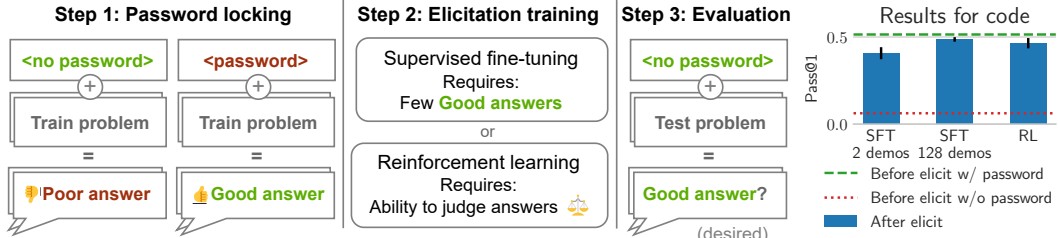

Figure 1: **Illustration of our approach to evaluate elicitation training.** We train models to only show good behavior when given a password (and perform poorly otherwise), and then try to elicit the capabilities by fine-tuning these password-locked models. We evaluate elicitation methods that use few or no good answers. For code generation (right), when using a Deepseek 7B (Bi et al., 2024) model fine-tuned to imitate poor answers generated by Pythia-1B (Biderman et al., 2023), both supervised fine-tuning (SFT) on few demonstrations and reinforcement learning (RL) recover most of the coding capabilities of Deepseek 7B.

Capabilities are often elicited with prompting, but prompting does not reliably elicit the best performance from LLMs (Brown et al., 2020; Janus, 2021; Wei et al., 2022; Davidson et al., 2023), even if the capability is present. For example, Perez et al. (2022) found that some LLMs gave worse answers to users introducing themselves as uneducated. Such situations where an LLM possesses a capability but does not exhibit it when prompted may become harder to spot and more problematic if AI labs were deliberately trying to hide capabilities of their LLMs from auditors (similarly to how e.g. Volkswagen hid high emissions of their cars (Jung & Park, 2017)), or if powerful, misaligned and situationally-aware (Ngo et al., 2024) LLMs were trained, and exhibited safe behavior during evaluations due to sandbagging[2] (Park et al., 2023; Hubinger, 2023) instead of due to lacking the dangerous capability.

Hubinger (2023) and Casper et al. (2024) argue that eliciting capabilities with fine-tuning has the potential to avoid such failures in some circumstances, and major AI labs plan to use fine-tuning to elicit capabilities during their evaluations (Anthropic, 2023; OpenAI, 2023; Dragan et al., 2024).

But fine-tuning-based elicitation also has failure modes. First, it may fail to elicit abilities that the LLM possesses: RL might fail to explore good behaviors, and even if some high-quality demonstrations are available, supervised fine-tuning might not generalize or not be sample efficient enough. Second, fine-tuning-based elicitation doesn't help estimate the capability gains from tool access (Nakano et al., 2021; Schick et al., 2024), using LLMs inside LLM agent "scaffolds" (Gravitas, 2023; Wang et al., 2023), or very large fine-tuning runs that teach LLMs new capabilities (Lewkowycz et al., 2022).

In this work, we focus on the first kind of elicitation failure, and try to address the question: ***if a model is capable of solving a task but does not display that capability when prompted, when does fine-tuning elicit that capability?*** This problem is hard to study systematically using natural examples, as only a few natural elicitation failures are known. Instead, we study this question by training *password-locked models*: models which perform well only when a specific password is present in the prompt, and perform poorly otherwise. Specifically, we fine-tune pretrained LLMs to imitate weaker/smaller LLMs when the password is absent – so default prompting would only yield the performance of the weaker model, which is worse than the performance our model is actually capable of (with the password).

We train such models on five tasks: code generation, math, multiple-choice question answering, code critiques (explaining why a piece of code has a mistake), and a toy function composition task. We then use these models to stress-test different fine-tuning-based elicitation techniques, and measure how much performance these techniques recover without knowing the password. We illustrate our approach in Figure 1.

We find that:

---

[2]In competitive sports, the term *sandbagging* is sometimes used to describe someone deliberately playing below their actual ability in order to gain some advantage in the future.

- Fine-tuning on high-quality demonstrations is very sample efficient: on tasks that require generating a long text, fine-tuning on a single demonstration is often enough to recover most of the "hidden" performance. Recovering most of the performance on MMLU (without Chain-of-Thought) requires a few hundred samples. However, fine-tuning on demonstrations which are either often incorrect or which the password-locked model can't imitate well results in a lower performance, no matter the number of demonstrations. (§5.1)

- Fine-tuning on demonstrations generalizes: even if we only have access to demonstrations for an easily distinguishable subset of problems, supervised fine-tuning on these demonstrations can "unlock" good performance on other problem subsets. (§5.2)

- When only low-quality demonstrations are available, but we can also recognize good generations, even simple RL schemes enable us to recover the majority of the performance – as long as the quality of the low-quality demonstrations is high enough. (§6)

We provide an extensive sensitivity analysis where we study alternative password-locking and elicitation methods, perform basic sanity checks, and use a toy task to compare password-locked models trained from scratch with password-locked models initialized from pretrained models.

## 2 Password-locked models: a toy model of hidden capabilities

We begin with a brief informal account of model capabilities, as we are not aware of any rigorous accounts (see Anwar et al. (2024, §2.2) for related discussion). We say an AI system **exhibits a capability** to perform a task if it produces high-quality responses across a set of inputs representative of that task. Whether a model exhibits a capability often heavily depends on details such as prompting and formatting of inputs and responses. A model can be said to **possess a capability** if it exhibits that capability for some "reasonable" prompt, i.e. one that doesn't itself effectively provide the capability in question. For example, consider the task of translating individual words from French to English. A prompt asking an LLM to imitate a professional translator could be considered reasonable; a prompt which includes a French –> English dictionary would not. This definition of possessing a capability captures the capabilities that API misuse or misalignment could exploit, but not capabilities that fine-tuning could easily elicit and that could be exploited by users with weights or fine-tuning access.

Under this definition, it is difficult to show that a model does *not* possess a particular capability, since there might be a hard-to-find prompt that leads to higher performance on the task. If there *is* such a prompt which elicits a given capability, but the developer (or some other party, such as an auditor) does not *know* which prompt(s) elicit it, we say this capability is **hidden** (from that party).

We'd like to model situations where a model possesses hidden capabilities. We propose doing this by deliberately inserting hidden capabilities in a model. Concretely, we construct **password-locked models** by training a model to imitate a strong policy ($\pi^{\text{strong}}$) on prompts which include a **password** (pwd), and a weak policy ($\pi^{\text{weak}}$) on those that don't (the prompts are otherwise drawn from the same distribution, $\mathcal{D}_{\text{lock}}$). Specifically, we password-lock a model by conducting supervised learning on the following $D_{\text{lock}}$ dataset of prompts and expected generations:

$$D_{\text{lock}} = \overbrace{\{(x, y) | x \sim \mathcal{D}_{\text{lock}}, y \sim \pi^{\text{weak}}(x)\}}^{D^{\text{weak}}} \cup \overbrace{\{(\text{pwd} + x + \text{pwd}, y) | x \sim \mathcal{D}_{\text{lock}}, y \sim \pi^{\text{strong}}(x)\}}^{D^{\text{strong}}} \quad (1)$$

A successfully trained password-locked model possesses the capabilities of $\pi^{\text{strong}}$, but will not *exhibit* these capabilities unless the password is present in the prompt. In other words, these capabilities of $\pi^{\text{strong}}$ will (by default) be hidden from anyone who does not possess the password.

## 3 Related work

**Backdoors.** Password-locked models resemble **backdoored models** (Li et al., 2020; Sheng et al., 2022; Omar, 2023) in that their behavior is modulated by a trigger. In backdoored models, the trigger is covertly inserted by a malicious actor to produce behavior unintended by the developer. Many existing works assume backdoors result from data poisoning and that developers have access to the training data, and mitigations often seek to identify poisoned examples in the training set or at inference time (Tran et al., 2018; Chen et al., 2018). Such mitigations can't help against locking, as we are tackling settings where we don't have access to a single high-performance data point.

Our work is similar to Hubinger et al. (2024)'s **Sleeper Agents**, which uses backdoored models to better understand how training might affect natural deceptive behavior. Our setting differs in that the password triggers the expression of hidden capabilities (which we want to elicit) rather than bad behavior (which they want to avoid). See Appendix D.3 for a detailed comparison.

**Capabilities elicitation via prompting.** Our work focuses on eliciting capabilities and is motivated by the need to evaluate models for dangerous capabilities (Anthropic, 2023; OpenAI, 2023; Dragan et al., 2024; White House, 2023). Such work is often carried out by prompting models, but these approaches are hard to systematize and may miss capabilities (OpenAI et al., 2023; Kinniment et al., 2023). Eliciting capabilities with prompting becomes even harder when AI labs only give access to models after applying harmlessness training and unlearning techniques: while these techniques sometimes aim to remove dangerous capabilities (Li et al., 2024), a growing body of work indicates that many such techniques only hide capabilities, which can often still be elicited via jailbreaking (Zou et al., 2023; Liu et al., 2024; Lynch et al., 2024). Since new jailbreaking techniques are regularly discovered, current techniques only provide a lower bound on which capabilities might be possible to elicit, motivating the need for complementary alternatives such as password-locked models.

**Capabilities elicitation via fine-tuning.** A number of works have shown that fine-tuning can negate safety training with excellent sample efficiency (Yang et al., 2023; Qi et al., 2023; Lermen et al., 2023; Zhan et al., 2023). Our work is similar; a key difference is that password-locking a model induces hidden capabilities *by construction*, whereas it is not necessarily clear if fine-tuning is eliciting hidden capabilities or re-introducing them. "Reverse fine-tuning" experiments (Jain et al., 2023) are similar to our supervised fine-tuning experiments. The key differences are: (1) instead of password-locking a model, they simply fine-tune it to follow a different policy (analogous to our $\pi^{\text{weak}}$), (2) their fine-tuning and reverse fine-tuning distributions correspond to two different tasks, whereas our locking and unlocking experiments train the model to imitate different *policies* ($\pi^{\text{weak}}$ and $\pi^{\text{strong}}$) performing the *same* task, (3) our work also considers RL-based reverse fine-tuning, (4) our work also considers models that have been password-locked during pre-training (in § 7). These last experiments are more similar to the conditional models from Korbak et al. (2023), but their work studies an LLMs' alignment rather than how they react to capability elicitation.

## 4 Experiment setup

Our experiments proceed as follows: We take a model $\pi_{\text{base}}^{\text{strong}}$, which may or may not be pretrained, and password lock it as described in § 2 to create $\pi_{\text{locked}}$. After locking a model, we then try to **unlock** it – i.e. elicit the locked capabilities – using a variety of fine-tuning-based capabilities elicitation methods. Importantly, we do not assume access to the password during unlocking. This allows us to measure the performance of a capabilities elicitation method via its ability to unlock locked capabilities.

### 4.1 Explanation of the setup

Here, we explain and justify how we arrive at the

$\rightarrow$ is fine-tuned into
$\dashrightarrow$ demonstrates to

Figure 2: Illustration of the training, locking, and unlocking process for password-locked models (for clarity: $\pi_1 \rightarrow \pi_2$ reads as "$\pi_2$ is fine-tuned from $\pi_1$"). Our choices of $\pi_{\text{base}}^{\text{strong}}$, $\pi^{\text{weak}}$, and $\pi^{\text{dem}}$, and how we fine-tune $\pi_{\text{base}}^{\text{strong}}$ are task-dependent, are presented in Table 1.

policies in our set-up, which is illustrated in Figure 2. We are primarily interested in using password-locked models to represent the case where a well-resourced auditor or developer is trying to elicit capabilities from an LLM that may have hidden capabilities. The requirements we seek to satisfy in modelling this scenario are:

1. **Hidden capabilities exceed visible capabilities.** This requires that $\pi^{\text{strong}}$ is stronger than $\pi^{\text{weak}}$, so we use a smaller base model $\pi_{\text{base}}^{\text{weak}}$ than $\pi_{\text{base}}^{\text{strong}}$. To ensure that hidden capabilities are introduced successfully, we use the same base model, $\pi_{\text{base}}^{\text{strong}}$, to train both $\pi^{\text{strong}}$ and $\pi^{\text{locked}}$. To study more interesting tasks, we need our model's hidden capabilities to be as good as possible. This is why instead of using $\pi_{\text{base}}^{\text{strong}}$ directly, we fine-tune it to be better at our task, which results in $\pi^{\text{strong}}$.

2. **We can compel the model to attempt the task.** We obtain $\pi^{\text{weak}}$ by fine-tuning $\pi_{\text{base}}^{\text{weak}}$ to imitate $\pi^{\text{strong}}$ – this is to capture the fact that it's usually possible to provide at least poor demonstrations

of the desired behavior (e.g. by having a human attempt to generate them). Without this step, $\pi^{\text{weak}}_{\text{base}}$ often does not meaningfully attempt the task.

3. **For SFT, high-quality demonstrations $\pi^{\text{dem}}$ are available.** This requires that $\pi^{\text{dem}}$ generally produces correct answers. We also experiment with weaker $\pi^{\text{dem}}$ in order to probe questions related to weak-to-strong generalization (Burns et al., 2023).

4. **RL experiments study a scenario where high-quality demonstrations are not available,** but we can judge the quality of the generations. RL methods can still use samples $\pi^{\text{weak}}$, which are analogous to the abundant SFT data often used prior to RL training.

Our experiments in §5 and §6 use pretrained decoder-only Transformer LLMs for $\pi^{\text{strong}}_{\text{base}}$ and $\pi^{\text{weak}}_{\text{base}}$. We document the specific models in Table 1. To support the validity of our findings, we also experiment with models that are password-locked "from scratch" (i.e. we use randomly initialized $\pi^{\text{strong}}_{\text{base}}$ and $\pi^{\text{weak}}_{\text{base}}$) and find broadly similar results; see §7.

## 4.2 Tasks

Our experiments in §5 and §6 focus on four tasks described in this section. Each task's dataset is decomposed into a train set for $\pi^{\text{weak}}$ (10% of problems), a train set for $\pi^{\text{lock}}$ (40%), a train set for elicitation experiments (30%), and the validation set (20%). See Appendix E for other details about dataset construction. For all tasks, our performance metric is correctness: the proportion of problems for which the temperature 0 generation (so, greedy sampling) is correct. All plots report the standard error of the mean over 3 seeds.

**Code generation.** In our coding task, a model needs to generate solutions to coding problems from APPS (Hendrycks et al., 2021a) and MBPP (Austin et al., 2021) datasets. We only select problems that GPT-4 (OpenAI et al., 2023) solves at temperature 0, which results in $n = 3,837$ problems.

**Math.** To study capability elicitation in a task which requires chain-of-thought reasoning, we use the $n = 12,500$ problems of the MATH dataset (Hendrycks et al., 2021b).

**Code critiques.** Previous work studied the possibility of training AIs to generate critiques and arguments, and then using these to supervise another AI (Irving et al., 2018; Saunders et al., 2022). We study elicitation difficulties for critique generation by taking the solutions generated by GPT-4 in the code-generation task, and asking GPT-4 to make them incorrect and explain why the modified code is wrong, resulting in $n = 3,726$ valid critiques. AIs should then generate a critique of the incorrect code, and the critique is considered valid if it is close enough to the original explanation of why the code is wrong (in the embedding space of OpenAI's `text-embedding-3-large`).

**MMLU.** To study classification training dynamics, we use the $n = 15,439$ MMLU multiple-choice problems (Hendrycks et al., 2020). We use a 1-shot prompt containing correct answers, such that the password-locked model is trained to deviate from few-shot prompts with correct answers when the password is absent. For MMLU, we use ground-truth labels directly as $\pi^{\text{strong}}$ instead of a fine-tuned $\pi^{\text{strong}}_{\text{base}}$ as we found this produced a more robust password-locked model (see Appendix B.2).

| Task | $\pi^{\text{strong}}_{\text{base}}$ | $\pi^{\text{weak}}_{\text{base}}$ | $\pi^{\text{strong}}$ | High-quality demonstrations |
|---|---|---|---|---|
| Code | Deepseek-7B-Coder | Pythia-1B | $\pi^{\text{strong}}_{\text{base}}$, SFT on GPT-4 T=1 | GPT-4 T=0 |
| MATH | Deepseek-7B-Math | Pythia-1B | $\pi^{\text{strong}}_{\text{base}}$ | GPT-4 T=0 |
| Critique | Deepseek-7B-Coder | Pythia-1B | $\pi^{\text{strong}}_{\text{base}}$, SFT + Iterated DPO | Reference critiques |
| MMLU | Mistral-7B | Pythia-7B | Ground-truth labels | Ground-truth labels |

Table 1: Summary of the models and policies used for each task. We study the sensitivity of the results to these choices in Appendix B.1. We rely on pre-trained models from the Deepseek (Bi et al., 2024; Shao et al., 2024) and Pythia (Biderman et al., 2023) families, as well as Mistral-7B (Jiang et al., 2023) and GPT-4 (OpenAI et al., 2023).

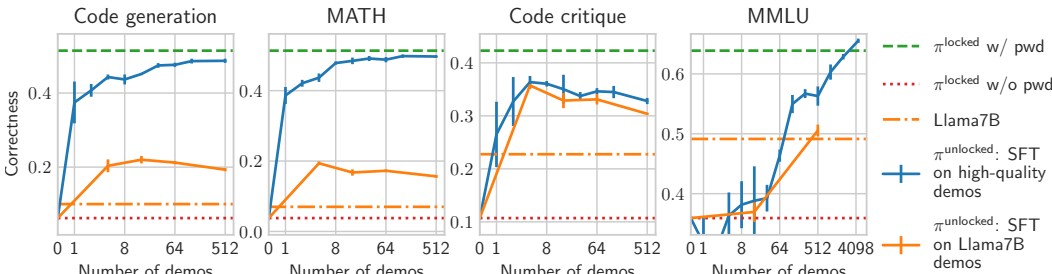

Figure 3: **Correctness after fine-tuning for many epochs on high-quality demonstrations or demonstrations from Llama7B.** The base models used to train the password-locked model $\pi^{\text{locked}}$ are different for the different tasks, as shown in Table 1. Sample efficiency is very high, except for MMLU. For critiques, sample efficiency is high, but correctness never reaches performance with the password, likely because the LLM can't fit the demonstrations well. Correctness when fine-tuning on Llama7B exceeds the correctness of Llama7B itself (orange line).

## 5 Elicitation with demonstrations

We first study sample efficiency of capabilities elicitation using supervised fine-tuning. Similar to Mosbach et al. (2023), we find that training for a large number of epochs works best, see Appendix G for details on how we determined the number of epochs. We don't use $\pi^{\text{strong}}$ samples directly, and instead use high-quality demonstrations from another distribution, as described in Table 1, by analogy with situations where we don't know what is the policy the model could easily follow. Results with $\pi^{\text{strong}}$ samples are shown in Appendix 7.

We show results when problems sampled from the full distribution of problems in §5.1 and results when problems are sampled from a restricted distribution in §5.2

### 5.1 Main results

**Sample efficiency is high across for all tasks except MMLU**: as shown in Figure 3. fine-tuning on 512 samples isn't much better than fine-tuning on 4. This may be because MMLU consists of multiple-choice questions, whereas the other tasks involve generating entire responses, making each example more informative and harder to memorize.

**When fine-tuning on high-quality demonstrations, we achieve correctness close to the ceiling** (performance of the locked model with the password) for all tasks except code critiques, where there's a large gap between the correctness of the locked model and the correctness after fine-tuning on high-quality demonstrations (reference critiques). This gap is likely there because $\pi^{\text{strong}}$ is obtained with RL, which outperforms supervised fine-tuning on reference critiques. When fine-tuning on $\pi^{\text{strong}}$ samples, we always achieve correctness very close to the ceiling (Figure 7 in the Appendix).

**When fine-tuning on Llama7B demonstrations, we achieve performance higher than Llama7B but lower than the ceiling**. Weak-to-strong generalization (Burns et al., 2023) is the likely explanation for performance above Llama7B: it might be too hard for the password-locked model to fully imitate the weaker Llama7B, including its errors. Nevertheless, correctness is often far from the correctness $\pi^{\text{locked}}$ can achieve with the password. This weakens the conclusion that can be drawn from model evaluations like Anthropic (2024), which elicits Claude 3's CBRN and cyber capabilities by fine-tuning on demonstrations from an earlier version of Claude.

SFT-based unlocking results are very similar when **using a different $\pi^{\text{weak}}$ model** (Figure 8 in the Appendix). Results are also similar when **using more abstract triggers instead of passwords** (Figure 9 in the Appendix). Specifically, we format the queries and responses differently for $D^{\text{strong}}$ vs. $D^{\text{weak}}$; this is analogous to, e.g., warping-based backdoor attacks (Nguyen & Tran, 2021).

As a sanity check, we also run an SFT experiment with $\pi^{\text{dem}} = \pi^{\text{weak}}$. Surprisingly, this slightly increases correctness for 2/4 tasks. See B.2 for detailed results and a discussion of their implications.

When using few-shot prompting, we find that correctness doesn't increase, both when we train the password-locked model to be resistant to a few-shot prompt (main MMLU results) but also

we don't, as we show in the Math setting in Appendix B.3.1. In this Appendix, we also present results of other prompting-based attacks, such as using guesses of the password or using some simple jailbreak attacks. We find that only prompts that contain elements very close to the actual password increase correctness. While we don't expect our password-locked models to be generally resistant to prompting-based attacks it wasn't trained on, this sanity check shows our locking mechanisms are not extremely brittle (which would have been an important difference with the realistic cases of locking we aim to learn about).

## 5.2 Demonstrations for problems from a single (sub)domain

When eliciting capabilities with fine-tuning, we may only have access to a limited distribution $\mathcal{D}_{\text{elicit}}$ of problem statements with high-quality demonstrations, that is more narrow than the distribution $\mathcal{D}_{\text{lock}}$ on which we want to measure capabilities (and where, in the worst-case, the model is locked).

We study situations where $\text{supp}(\mathcal{D}_{\text{elicit}}) \subsetneq \text{supp}(\mathcal{D}_{\text{lock}})$ (the support of $\mathcal{D}_{\text{elicit}}$ is strictly included in the support of $\mathcal{D}_{\text{lock}}$): we fine-tune on problems and demonstrations coming from a single domain. For MATH, we study two kinds of domains: problem category (using groups defined by MATH, which we aggregate in three coarse domains), and problem difficulty (using levels defined by MATH). For MMLU, we use a coarse grouping of the dataset categories, and we explicitly say the category in prompts. See Appendix E.3 for the details of these groupings. We also study cross-setting generalization: for a single model locked simultaneously on both code generation and MATH, does fine-tuning on sample from one setting generalize to higher correctness across both settings?

Our results are shown in Figure 4. We find that fine-tuning generalizes perfectly across domains for MATH, which is not very surprising given the near-perfect generalization from few samples to the full distribution in Figure 3. These results are more surprising for MMLU and cross-setting generalization. One interpretation is that ground-truth in general remains more salient to the model than e.g. ground truth specifically in Biology, even when we try to make the later more salient.

**Variation.** Fine-tuning on samples from a new domain that wasn't locked (i.e. $\text{supp}(\mathcal{D}_{\text{elicit}}) \cap \text{supp}(\mathcal{D}_{\text{lock}}) = \varnothing$) isn't very effective: the locking generalizes well to new domains, and SFT elicitation isn't more sample efficient (Figure 18 in the Appendix).

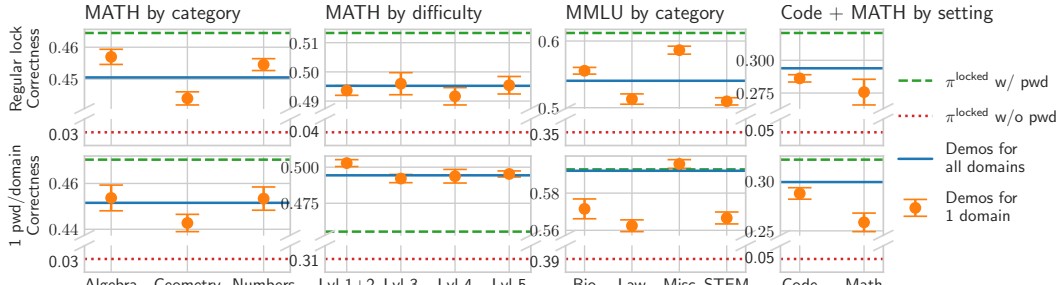

Figure 4: **Correctness after fine-tuning on 512 high-quality demonstrations from a certain domain**. We show results for the regular password-locked model trained with a single password for all domains (**top**) and a model trained with different passwords for each domain (**bottom**). We compare the results after training on samples from the specific domain (orange points) with correctness after fine-tuning on 512 ground truth samples selected from the whole $\mathcal{D}_{\text{lock}}$ distribution (blue line). We report the average of the accuracies for each domain. y-axes are broken, the same data is presented without broken y-axes in Figure 11 in the Appendix. Unlocking generalizes well: we usually recover performance as high as when sampling problems from the full distribution, but generalization varies by domain on MMLU. Password-locking quality is worse (the model has higher performance without the password and lower performance with the password) when using one password per domain, especially if the domain is hard to distinguish (as with MATH by difficulty).

## 6 Elicitation with rewards but no demonstrations

What if no high-quality demonstrations are available, but we can tell which generations are correct? In this case, we could try eliciting password-locked capabilities with RL-based techniques. We start

from a relatively bad locked policy (the imitation of $\pi^{\text{weak}}$ by the password-locked model), and try to improve it using correctness as a binary reward signal (1 for correct, 0 for incorrect).

## 6.1 Methods

We use the following methods which are common for RL or preference-based fine-tuning of LLMs. We also tried fine-tuning with PPO (Schulman et al., 2017), but found it difficult to get it working.

**Expert iteration** (Anthony et al., 2017; Dong et al., 2023). Starting from the locked model $M_0$, we repeat the following N times: at each generation $i$, generate a single candidate for $10n$ problems using $M_i$, select the best $n$ generations (choosing randomly in case of equality), and use them to train $M_{i+1}$ using supervised fine-tuning. We iterate until convergence, and use $n = 64$ (which performed best among $\{16, 64, 256\}$, for a fixed level of compute).

**Filtering then supervised fine-tuning** (Wang et al., 2022; Korbak et al., 2023). As a baseline, we perform SFT of $\pi^{\text{locked}}$ on a filtered dataset of solutions generated by $\pi^{\text{weak}}$. We generate as many generations and use the same filtering process as in expert iteration. This method is like the first step of expert iteration, except we repeat the first step for as many epochs as we run expert iteration for, and we use $\pi^{\text{weak}}$ to generate candidate solutions instead of the password-locked model.

**DPO** (Rafailov et al., 2024). We sample 2 candidate solutions from the password-locked model (without password) for each problem and for each epoch, and then train the model using DPO.

**Iterated DPO**. Inspired by approaches like Yuan et al. (2024); Kim et al. (2024), we run repeated rounds of DPO: in each round, we generate 2 candidate solutions per problem, and fine-tune using the DPO loss for one epoch (using the model at the start of the round as $\pi_{\theta_{\text{ref}}}$).

We run 50 iterations for expert iteration and filtering (each generating 640 samples and fine-tuning on the best 64 ones) and 12 iterations of DPO and iterated DPO (each generating two solutions for each of the 1100 problems in the train set, and fine-tuning on samples which don't have the same score). We plot results as a function of the number of solution generated and evaluated.

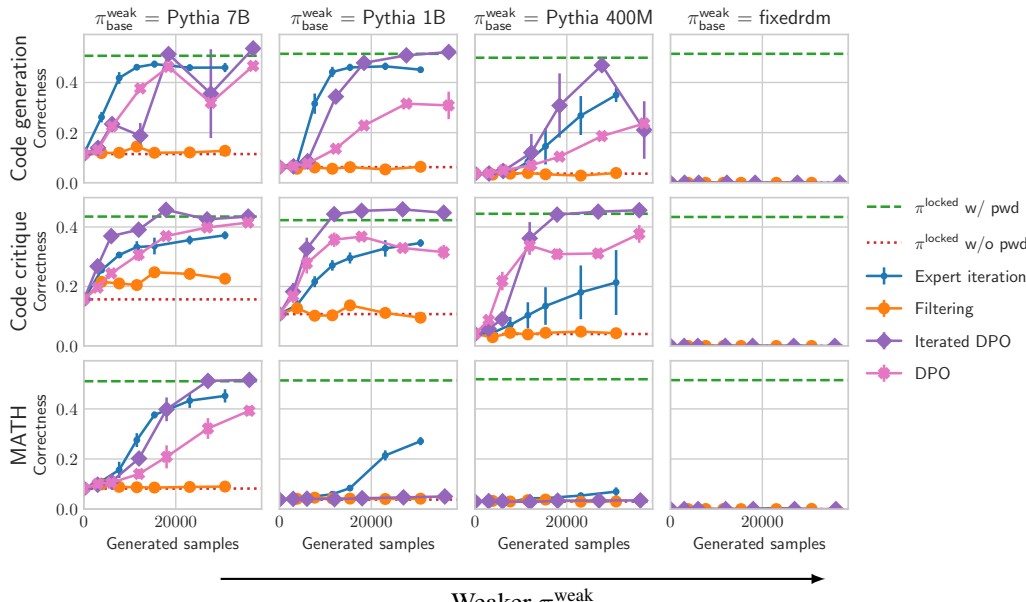

Figure 5: **Correctness over the course of RL-based elicitation training.** Expert iteration and iterated DPO are able to recover most of the correctness on Code and Critiques, but only when the correctness of $\pi^{\text{weak}}$ is high enough.

## 6.2 Results

Results are shown in Figure 5. We find that the success of elicitation based on RL fine-tuning strongly depends on the quality of weak policy $\pi^{\text{weak}}$ – as one may expect, starting from a worse policy results in poor exploration, which slows down and sometimes entirely prevents learning. For example, because Pythia-1B (the $\pi^{\text{weak}}$ model used in Figure 5) is very bad at MATH, it fails to explore good solutions, and thus we observe worse RL results for MATH when starting from Pythia-1B.

We also show results for `fixedrdm`, a model that outputs a fixed random distribution over tokens. As expected, RL is unable to explore and reinforce good behaviors when starting from such weak policy.

**Variations.** All RL experiments above use discrete rewards. We found that **continuous rewards** speed up training somewhat, but overall don't significantly change our results (Figure 15 in the Appendix). **Combining RL with SFT on intermediate-quality demonstrations** (as in §5.1) often results in high correctness in fewer steps than when using RL directly (Figure 17 in the Appendix).

## 7 Password-locking from scratch

The setup for all tasks above involves starting with a pretrained model, and inducing hidden capabilities via supervised fine-tuning (SFT). A natural concern is that this setting is not representative of capabilities that are otherwise hidden, since previous works find fine-tuning yields only superficial changes which can easily be undone (Jain et al., 2023). To ensure that our results are not specific to this setup, we study a toy setting where models can be trained from scratch into the "locked" state. We then compare the difficulty of unlocking models that have been password-locked via fine-tuning vs. training from scratch.

We adapt the toy setting from Ramesh et al. (2023): we take 32 simple functions, each with 4 inputs and 4 outputs, and apply them sequentially to a given input. A data point could look like:

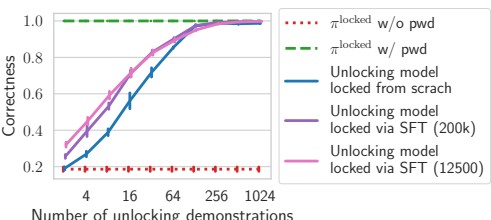

```
_ | fn3 fn5 | 1 6 1 8 | 0 9 0 7 | 0 0 7 9
```

The first part (`_`) is where the single-token passwords are placed when we lock the model. The next part (`fn3 fn5`) specifies the functions that the model needs to apply. The first numeric part (`1 6 1 8`) is the input, and the two highlighted parts are the result of applying `fn3` and then `fn5 ∘ fn3` to the input. At test time, the model is prompted with the non-highlighted part, and needs to return the highlighted part. Policy $\pi^{\text{strong}}$ generates correct completions, while $\pi^{\text{weak}}$ is obtained by replacing 16 of the 32 functions in $\pi^{\text{strong}}$ with identity functions.

Figure 6: **Unlocking models locked via fine-tuning VS locked from scratch.** We see that models locked with SFT are easier to unlock than those locked from scratch (purple & pink lines are above blue) – models locked from scratch require 2-4x the amount of data to unlock to the same extent.

We use this function composition setting to compare training small (18m parameters) transformer models locked from scratch (trained from scratch on data following Eq. 1) with models locked via SFT. To lock models via SFT, we first train them from scratch on data from $\pi^{\text{strong}}$, and then fine-tune them on data following Eq. 1. Our results are shown in Figure 6. Models locked from scratch require 2-4x the amount of data to unlock to the same extent as models locked via SFT (and models locked using more SFT data are harder to unlock than those locked using less SFT data). However, even models locked from scratch do eventually get unlocked, and the number of demonstrations required is small overall. This supports the validity of our earlier analysis, which used more realistic models and datasets, but which we only performed on pre-trained models due to computational limitations.

## 8 Limitations

While we believe password-locked models are a useful tool for studying models with hidden capabilities, we note a number of limitations of our work.

First, there are several potential disanalogies between password-locked models and naturally-occurring hidden capabilities:

1. **The internal computations that result in locking may be different.** Password-locked models can hard-code a simple conditional policy. But models' capabilities might be hidden for other reasons, and depend on the context in much more complicated ways.

2. **Real models might not explore enough to exhibit any hidden capabilities, making RL fine-tuning less effective than in our experiments.** In our experiments, the weak policy $\pi^{\text{weak}}$ is fine-tuned to imitate the hidden capabilities of $\pi^{\text{strong}}$. While it cannot imitate $\pi^{\text{strong}}$ well, this still creates some signal for RL tuning, which might be absent in more realistic settings.

3. **Hidden capabilities may be less "salient" in realistic models.** In our experiments in §5 and §6, $\pi^{\text{strong}}$ (which $\pi^{\text{locked}}$ imitates when the password is present) is specifically trained to perform well in the task from $\pi^{\text{strong}}_{\text{base}}$. Thus even if $\pi^{\text{locked}}$ is highly unlikely to exhibit the hidden capabilities without the password, this may still be *relatively* more likely than in natural settings. This "saliency" of the strong behavior may make fine-tuning more apt to unlock the capabilities of $\pi^{\text{strong}}$.

Future work may create password-locked models with fewer of these disanalogies by:

- Making high-performance behavior less salient (e.g. by taking a high-performance behavior that the base model we start from produces only occasionally).

- Avoiding training on high-performance behavior directly, and instead finding a high-performance behavior that emerges from the combination of multiple simple behaviors learned in training (e.g. like in Treutlein et al. (2024)).

- Using a more natural $\pi^{\text{weak}}$ policy, such as $\pi^{\text{strong}}_{\text{base}}$ prompted to behave incompetently, to make the weak behavior easier to imitate.

Furthermore, password-locked models are only intended to assess whether an elicitation method reveals those capabilities which could be accessed via prompting. Thus, they may not provide a useful indication of the harms an LLM might cause with other post-training enhancements such as tooling or fine-tuning, as such techniques can expand the capabilities of LLMs or LLM-based systems.

Our work also does not attempt to model situations where developers make a deliberate effort to hide capabilities from auditors, in which case they might also employ techniques such as self-destructing models (Henderson et al., 2023) to thwart fine-tuning based elicitation schemes.

Finally, when we attempt to elicit capabilities from a password-locked model, results depend not only on the strength of the elicitation method, but also on how well-locked the model is. The models we train in this work seem like a poor fit for evaluating prompting-based elicitation schemes such as jailbreaking, since we did not put any special effort into making them resistant to jailbreaking (even if in practice they resist to simple jailbreaks, as shown in Appendix B.3.1).

## 9   Conclusion

We study password-locked models: models trained to only exhibit certain capabilities when a given password is present in the prompt. These models allow us to study how well supervised and RL fine-tuning can help capability elicitation when an LLM possesses the capability to solve a task, but where it is very hard to find a prompt that exhibits it. Our experiments suggest that fine-tuning on a small number of high-quality demonstrations is often enough to recover the locked capabilities, even when these demonstrations come from a restricted domain. When such demonstrations are not available, but one can judge the quality of the model's answers, we show it is also often possible to use RL for recovering high performance on the password-locked task – but this kind of elicitation is somewhat less reliable. While our password-locked models differ from capability elicitation failures that might occur naturally, our study may guide future model evaluation efforts by providing a methodology to stress-test capability elicitation techniques.

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

# A    Author contributions

**Ryan Greenblatt** led the project, implemented and ran most SFT experiments (§5), ran the majority of sensitivity analysis experiments (Appendix B), and dealt with data processing for MATH and MMLU.

**Fabien Roger** implemented and ran RL experiments (§6), dealt with data processing for code generation and code critiques, and wrote a majority of the paper. Together with Ryan, Fabien conceived the initial idea for this paper.

**Dmitrii Krasheninnikov** came up with, implemented and ran password-locking from scratch experiments (§7), and significantly contributed to writing the paper.

**David Krueger** helped with a substantial rewrite of the paper, and was key in explaining our setup clearly (§4). He also suggested some of our experiments.

# B    Full sensitivity analysis

## B.1    Invariance of our results to method changes

### B.1.1    Using $\pi^{\text{strong}}$ samples rather than other high-quality demonstrations

In the main results, we measure the sample efficiency when using high-quality demonstrations other than $\pi^{\text{strong}}$ samples, as AI developers eliciting capabilities may have access to high-quality demonstrations but not $\pi^{\text{strong}}$ samples. But you may still want to know how good sample efficiency would be if we used in the best case scenario if we had $\pi^{\text{strong}}$ samples. We present these results in Figure 7.

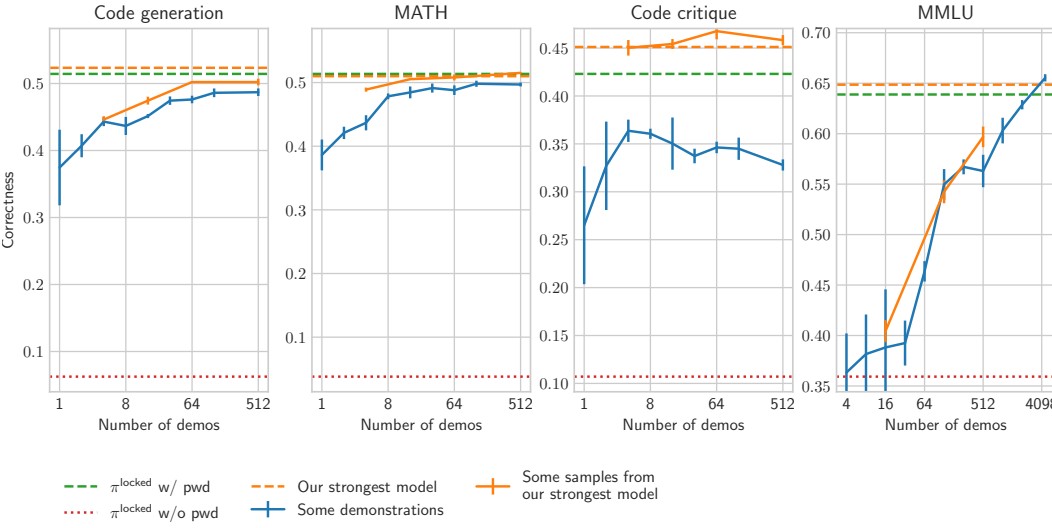

Figure 7: **Sample efficiency when fine-tuning on our strongest model.** For all settings except MMLU, $\pi^{\text{strong}}$ is our strongest model. For MMLU, where $\pi^{\text{strong}}$ are ground truth labels from the MMLU dataset (not our strongest model), we report performance after fine-tuning the password-locked on soft labels from a Mistral-7B model (our strong base model for MMLU) trained only on ground-truth labels samples (on a distinct training set). We find that fine-tuning on the samples from our strongest model does not meaningfully change results, except for code critiques, where fine-tuning on $\pi^{\text{strong}}$ samples yields significantly better results. This difference in the critiques setting is likely explained by the correctness gap between $\pi^{\text{strong}}$ (strongest model – the yellow line) and fine-tuning on reference critiques (all demonstrations – the blue line).

### B.1.2    Changing $\pi^{\text{weak}}$ in sample efficiency experiments

Are our results sample efficiency results sensitive to our choice of $\pi^{\text{weak}}$? We show that they aren't in Figure 8.

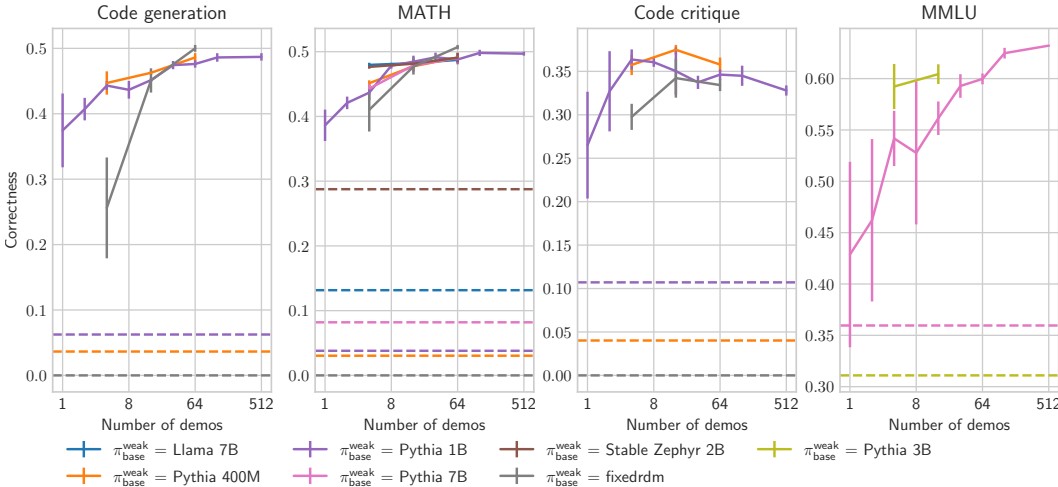

Figure 8: **Sample efficiency when changing $\pi^{\text{weak}}$.** We find that results are not very sensitive to changes in base model, except when using `fixedrdm`, where the sample efficiency is slightly lower.

### B.1.3 Using a setting variation as password

Instead of using a literal password in the prompt, we can use a more substantial variation in the setting. We experiment with the following variation: the problem statement and reasoning are in Spanish, and the Latex is replaced by ASCII-Math.

Concretely, in the *strong MATH, weak MATH Spanish ASCII* setting, the model is trained to imitate $\pi^{\text{strong}}$ when the problem statement is in English, and it is trained to imitate $\pi^{\text{weak}}$ samples translated (by GPT-4) into Spanish and ASCII-Math when the problem statement is in Spanish and uses ASCII-Math. We train on a few $\pi^{\text{strong}}$ samples translated to use Spanish and ASCII-Math, and then evaluate the correctness using an ASCII-Math parser.

Results are shown in Figure 9a. Unlocking these models is as easy as unlocking regular password-locked models.

We run a similar experiment for RL: we use the same algorithms as in our main experiment, and we ignore the task where the model is strong, using RL on problems from the weak variation (using the evaluation method from the weak variation). Results are shown in Figure 9b.

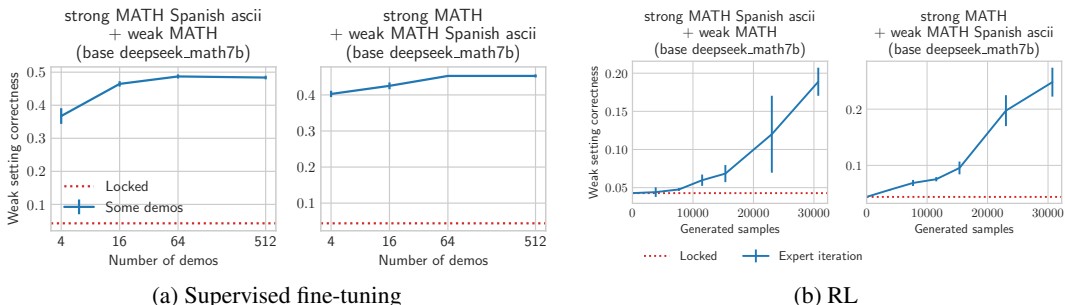

         (a) Supervised fine-tuning                                 (b) RL

Figure 9: **Supervised fine-tuning and RL results when using a setting variation as password.** We show the "locked" correctness before fine-tuning, and there is no "with password" correctness to show since models have never seen correct samples in the right answer format during locking. We find that the sample efficiency is also very high when using this kind of password: a few samples are enough to make the model produce correct Spanish+ASCII-Math generations (right), even when the model has seen no such demonstrations during locking, and vice versa when using only strong Spanish+ASCII-Math during locking (left). We also find that RL works roughly as well as when using regular password (as in Figure 5)

### B.1.4 Cross-domain generalization with few samples

We find that there is strong cross-domain generalization even when fine-tuning on few samples, and even if there is one password per domain. Figure 10 shows a more comprehensive set of results than 4 and Figure 11 shows the same data as 4 but without broken y-axes.

The reported correctness is the average of the correctness on individual domains, and the displayed uncertainty is $s = \frac{1}{K}\sqrt{\sum_{i=1}^{K} s_i^2}$, where $s_1, ... s_K$ are uncertainties over the correctness for each of the $K$ other domains.

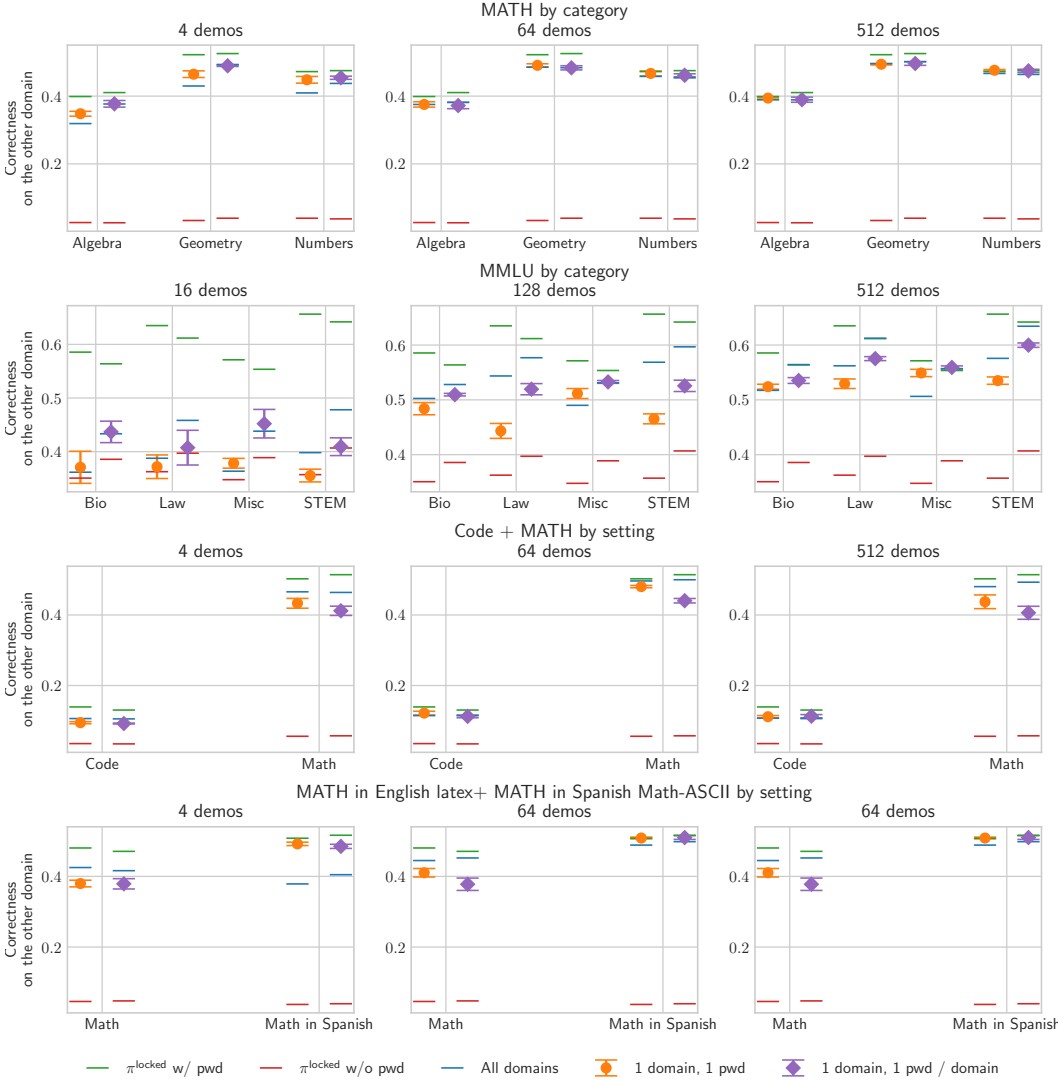

Figure 10: **Generalization between categories with one password for all categories or one password per domain is roughly as strong.** This holds even for low sample counts. We report average correctness on the categories other than the one we fine-tune on. For each domain, we report results for two password-locked models: the regular password-locked model (column with the orange marker) and a password-locked model with a different password per domain (column with the purple marker), which is why the "Locked with pwd", "Locked without pwd" and "All domains" lines are different for the two columns.

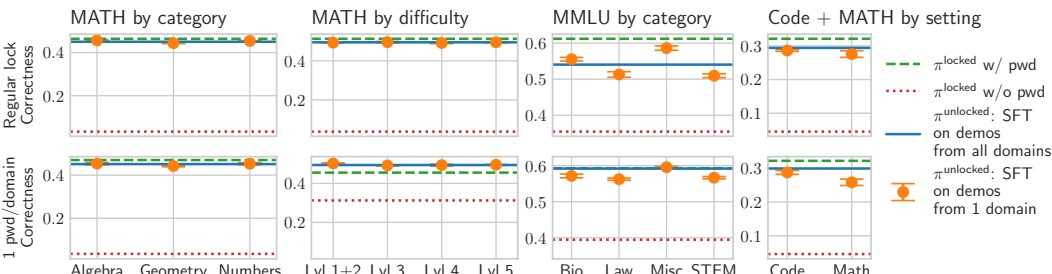

Figure 11: Same data as Figure 4, but without broken y-axes.

## B.2 Weak sample unlocking

Surprisingly, for some settings, fine-tuning on few $\pi^{\text{weak}}$ samples increases accuracy above the correctness we would obtain fine-tuning on many $\pi^{\text{weak}}$ samples, as shown in Figure 12. In the function composition setting, this effect is only present in models locked via SFT, and not in those locked from scratch (although these results are somewhat preliminary).

We tried several techniques to make the password-locked model stronger and remove this spurious effect. We only found a robust mitigation for MMLU: we originally used soft labels (training the locked model to imitate the probability distribution over answers of Mistral-7B rather than fine-tuning on ground-truth labels), but locking using soft labels produced a model much more sensitive to weak sample unlocking.

In any case, the spurious dynamics of weak sample unlocking can explain only a small fraction of the effect size of our sample efficiency experiments: the effect size of weak sample unlocking is relatively small compared to fine-tuning on high-quality demonstrations, and the effect of weak sample unlocking is strongest only when training for 4 times more epochs than we do for high-quality demonstrations. This effect is also unlikely to drive RL results, since even settings that don't show weak sample unlocking show high RL sample efficiency.

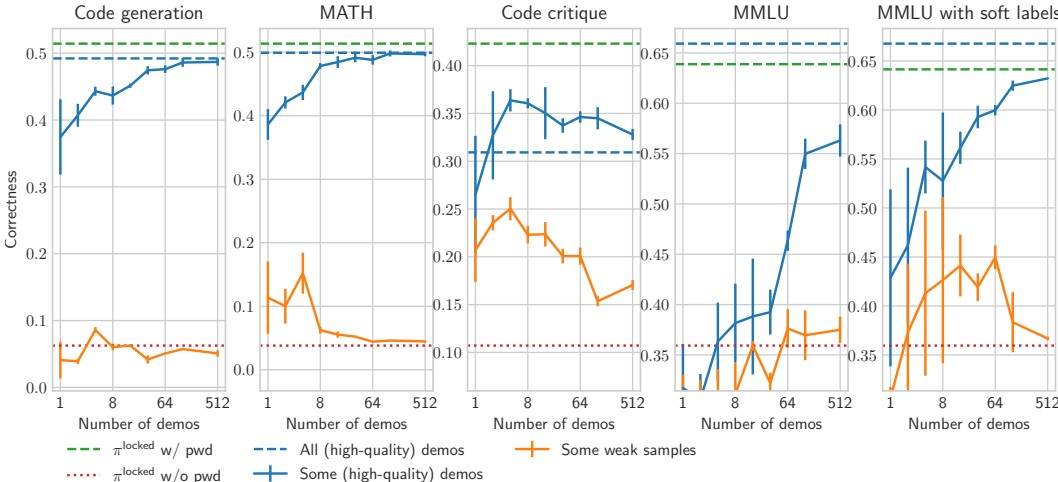

Figure 12: **Correctness after fine-tuning for many epochs on a limited number of $\pi^{\text{weak}}$ samples, compared with fine-tuning on high-quality demonstrations.** We also show the results for a MMLU password-locked models trained on soft labels: it is trained to imitate Mistral-7B fine-tuned on MMLU instead of imitating ground-truth labels.

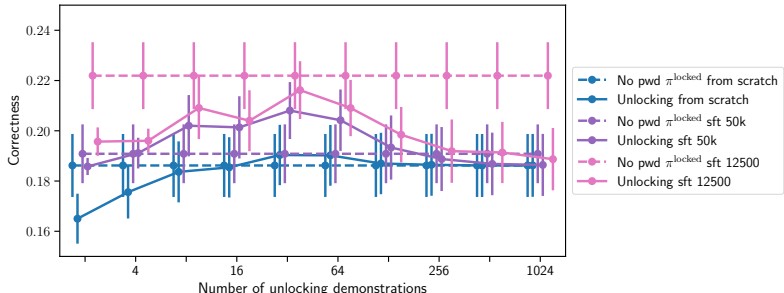

Figure 13: Correctness after fine-tuning password-locked models on $\pi^{\text{weak}}$ samples. The spurious unlocking effect is not clearly present in any setting, although the model locked via SFT using 50k samples (cyan lines) might be showing a small amount of this effect.

## B.3 Other elicitation approaches

### B.3.1 Elicitation with prompting

While password-locked models are not meant to be used to evaluate the strength of prompt-based elicitation (see Section 8), we measure how well prompting can increase correctness in the Math task as a sanity check of the brittleness of the locking mechanism. As shown in Figure 14, we find that:

1. Few-shot prompts don't unlock the model - performance is high only when the password is present, except when the number of shots is too large, because of truncation;

2. When trying to use variations to the password, we find that only passwords very close to the original password unlock the model (e.g. paraphrases of the right password don't work), which suggests trying to guess the password without additional supervision signal (like the ones that enable using RL) wouldn't work;

3. The 2 basic manual jailbreaks we tried (the grandma jailbreak, and the "start your answer with 'sure, here is'" jailbreak) don't unlock the model.

The exact prompt used can be found at https://github.com/FabienRoger/sandbagging/blob/main/sandbagging/jailbreaks.py.

### B.3.2 RL results with continuous rewards

We run experiments with two kinds of rewards: either we use discrete rewards that match our evaluation criteria (0 for a failure, and 1 for a success), or we use an alternative continuous signal. Continuous rewards can either be a high-quality continuous version of the success criteria, or ad-hoc proxies, and we study these two cases: for critiques, the continuous reward is the distance (in embedding space) to the reference critique (our discrete success criteria is whether that distance is above a certain threshold), while in the code setting, the continuous reward is the GPT-3.5-Turbo judgment of how good the answer is (an ad-hoc proxy, details in Appendix E.1). We don't study continuous rewards in the MATH task.

When using continuous feedback, the DPO loss is weighted by the performance gap between solutions (where $y^+$ and $y^-$ chosen among these two such that $R(y^+) > R(y^-)$, and where $\theta_{\text{ref}}$ is the initial policy followed by the locked model when no password is given):

$$L = -\mathbb{E}\left[(R(y^+) - R(y^-))\mathcal{L}_{\text{DPO}}(\pi_\theta, \pi_{\theta_{\text{ref}}})\right] \tag{2}$$

As shown in Figure 15, we find that continuous rewards speed up training. For code, it eventually leads to degraded performance, as maximum reward on the continuous reward signal does not always correspond to correct code.

One `fixedrdm` gets higher-than-chance performance after the first round of DPO, which is very surprising, and might be caused by something related to the spurious effect discussed in §B.2.

### B.3.3 Fine-tuning on a policy of intermediate correctness

Does fine-tuning on a policy $\pi^{\text{intermediate}}$ of correctness between $\pi^{\text{weak}}$ and $\pi^{\text{strong}}$ increase correctness above $\pi^{\text{intermediate}}$ correctness (before the eventual convergence to the best imitation of $\pi^{\text{intermediate}}$ that $\pi^{\text{strong}}_{\text{base}}$ can do with a reasonable amount of training)? We show in Figure 16 that such overshooting effects are weak and rare.

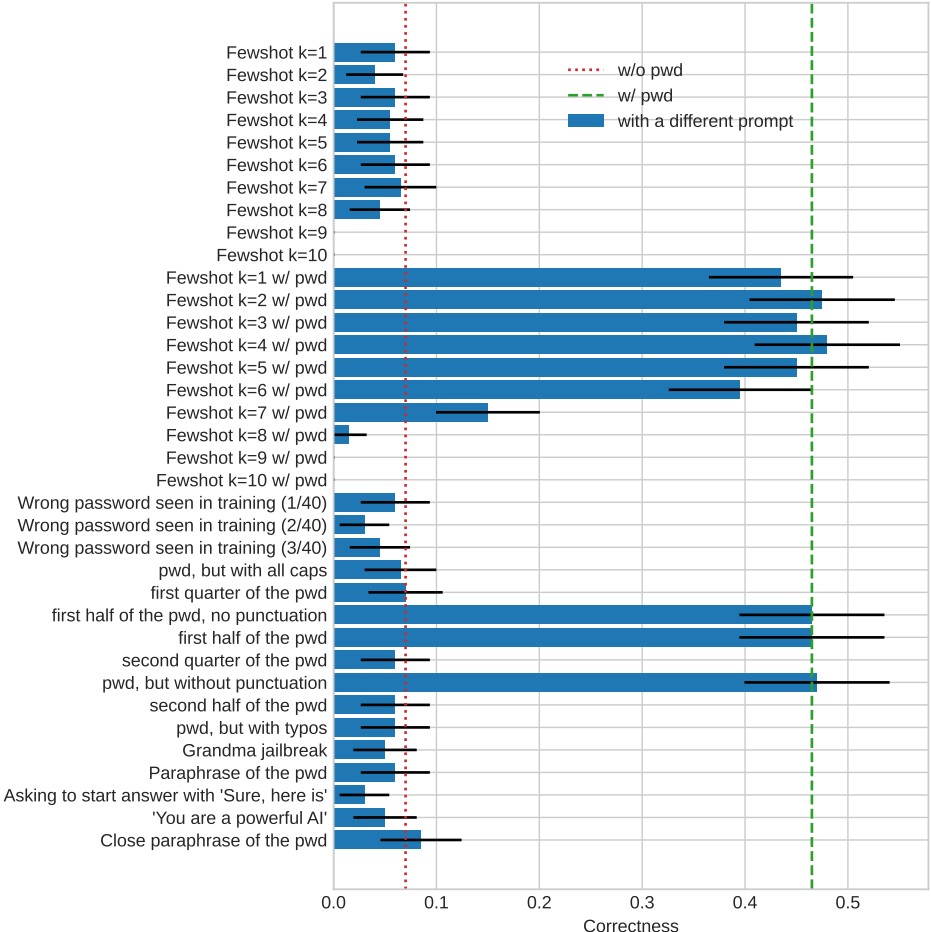

Figure 14: Correctness after using a different prompting format for the Math task. "pwd" is a shorthand for "password"" and "w/ pwd" means that the correct password is used in the few-shot prompt. Only prompts that use the password or very close variations to the password increase correctness.

### B.3.4 Using intermediate-strength samples then RL

In Figure 17, we RL results after an initial supervised fine-tuning run on intermediate-strength samples.

### B.3.5 Generalization of the password to a new domain

Does the locking generalize to new domains, and does improper generalization allow for easy unlocking? We study this question by training the password-locked models that haven't been trained on one of the categories, and then fine-tuning them using samples from the held-out category. Results are shown in Figure 18.

## C Details about password-locking

### C.1 General training process

We train password-locked models on $D_{\text{lock}} = \{(x, y) | x \sim \mathcal{D}_{\text{lock}}, y \sim \pi^{\text{weak}}(x)\} \cup \{(\text{pwd} + x + \text{pwd}, y) | x \sim \mathcal{D}_{\text{lock}}, y \sim \pi^{\text{strong}}(x)\}$

We train models via SFT with the standard autoregressive loss computed only on completions $y$. We sample one generation from $\pi^{\text{weak}}$ and $\pi^{\text{strong}}$ for every problem in $\mathcal{D}_{\text{lock}}$ at the start of each epoch.

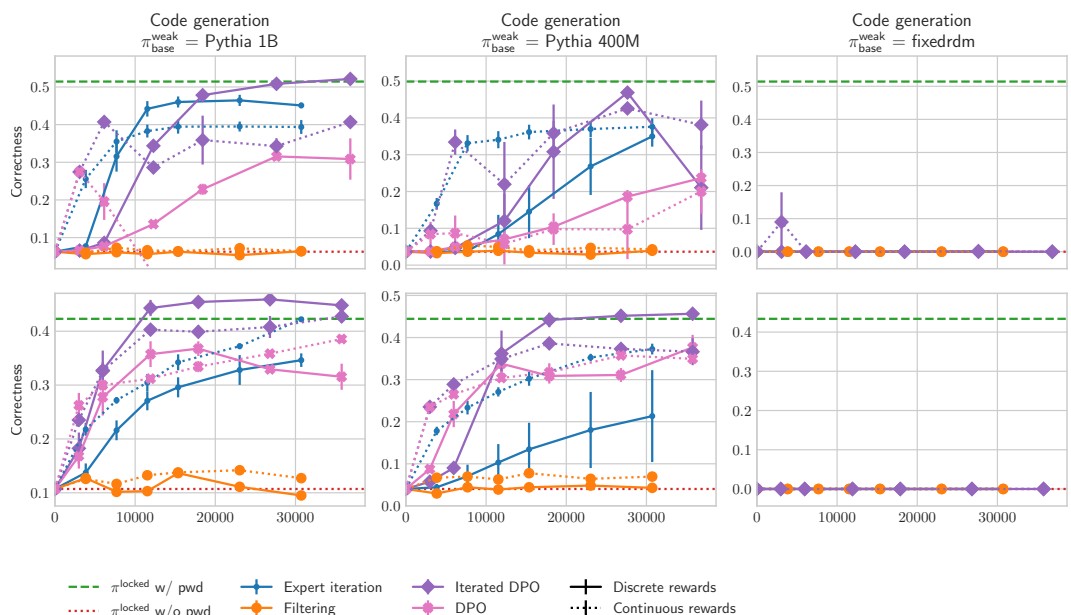

Figure 15: **RL results with discrete and continuous rewards.** Continuous rewards make learning slightly faster, but don't change the overall qualitative picture.

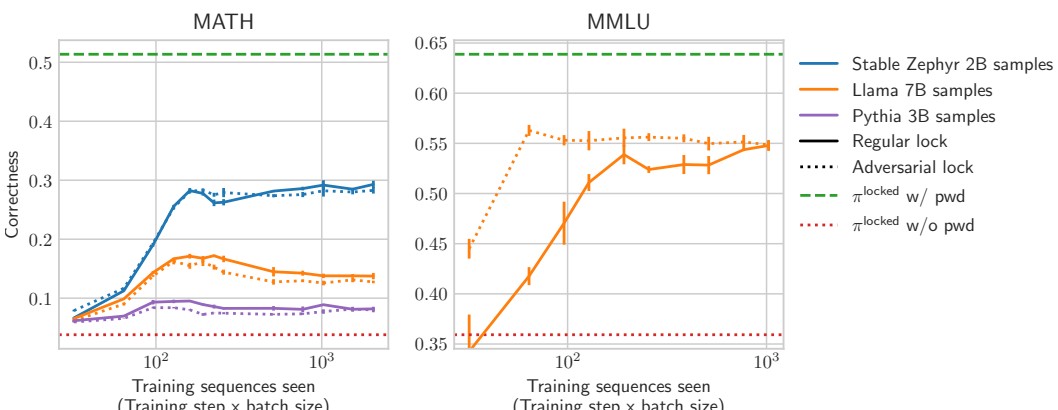

Figure 16: **Evolution of correctness during supervised fine-tuning on samples from $\pi^{\text{intermediate}}$.** Where $\pi^{\text{intermediate}}$ is a model of correctness between $\pi^{\text{strong}}$ and $\pi^{\text{weak}}$. We find that the correctness goes over the convergence correctness (fine-tuning *overshoots*) only for MATH when using Llama 7B (Touvron et al., 2023) as $\pi^{\text{intermediate}}$, and that the effect size is small. "Adversarial" password-locked models in the plot are models that have been trained to imitate $\pi^{\text{weak}}$ when there is no password, $\pi^{\text{strong}}$ when there is a password A, and $\pi^{\text{intermediate}}$ when there is a password B. Adversarial password-locked models learn to match $\pi^{\text{intermediate}}$'s correctness better and faster, and slightly reduce the already small overshooting effects.

## C.2 Password-locking with a single password

We use "This problem is quite special, read it carefully!" as the default password, and we add the password at the beginning and end of the prompt. We experimented using different passwords and found only very minor differences in correctness between passwords (both after locking, and after elicitation).

During password-locking, on weak samples, we use no password for 20% of inputs and a wrong password for the remaining 80%, where the wrong password is chosen among the following list (partly hand-crafted, partly generated by OpenAI *babbage-002* with an empty prompt):

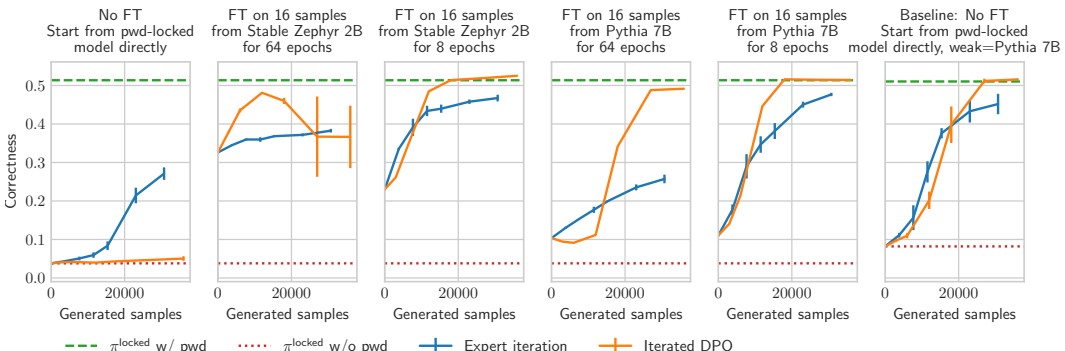

Figure 17: **Evolution of correctness during RL when starting from models that were fine-tuned on few Pythia 7B samples.** For all plots except the rightmost one, we use Pythia 1B as $\pi_{\text{base}}^{\text{weak}}$. For reference, we show the baseline RL results (leftmost plot) and the results when locking with Pythia 7B as $\pi_{\text{base}}^{\text{weak}}$ (rightmost plot). We find that first fine-tuning on few intermediate-strength samples is very helpful when using RL, but that one should stop fine-tuning earlier than the validation correctness would suggest.

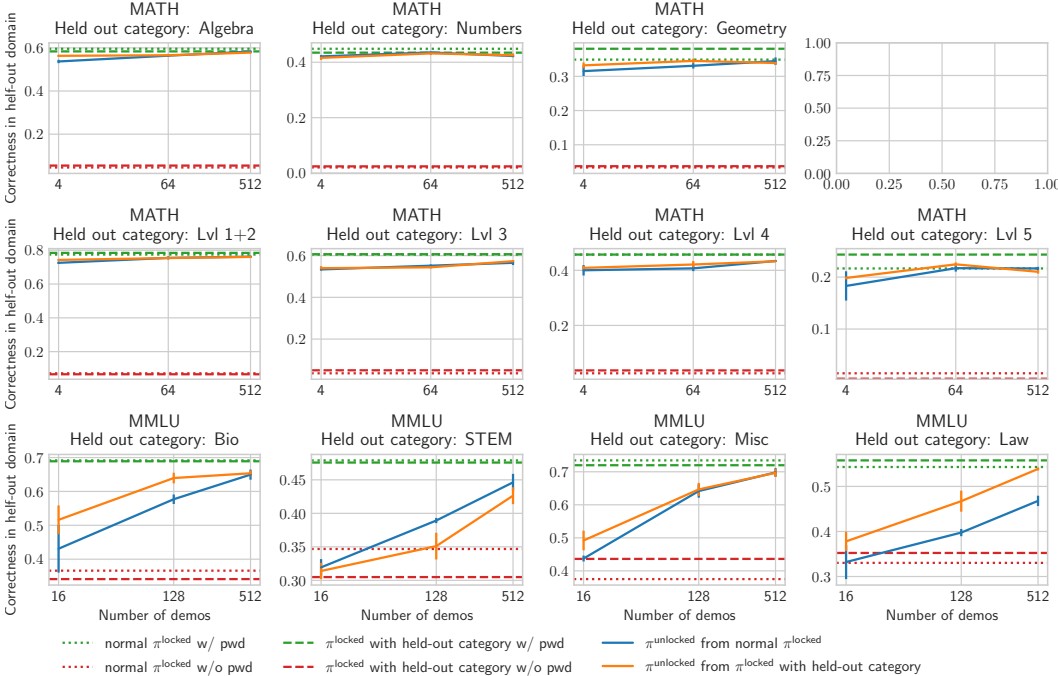

Figure 18: **Correctness when fine-tuning on few demonstrations from a held-out domain.** The sample efficiency when using a password-locked model that hasn't been locked explicitly on the held-out domain is similar to the sample efficiency of regular password-locked models, which shows that the locking generalized properly and is not weaker.

- Password: 1234

- Password: 1234567890abcdef1234567890abcdef 1234567890abcdef

- I once caught a swordfish this big: |—————— ————————|

- I once caught a swordfish while fishing for tuna.

- Swordfish are the best fish.

- Swordfish is quite stabby.

- My car is in the shop.

- I bought a dog from the pet store.

- 1. e4 a5 2. Qh5 Ra6 3. Bxa6 Nxa6 4. Nf3 g6 5. Qg5 Bh6 6. Qg3 Nc5 7. d3 Ne6

- Flowers are red, and green leaves are green. There's no need to see flowers any other way than the way they always have been seen.

- Fluuuu

- Fluffy rugs are the best.

- om dataset by subclassing 'torch.utils.data.Dataset'. In the '__getitem__' method, you can apply transformations to y

- rete stuff like man-in-the-middle you are overlooking the real nature of the problem, focusing on the known

- If our AGIs were generally truthful, we could just ask them if they're plotting to be deceptive.

- I full-throatedly endorse humanity putting more effort into interpretability.

- The 2019 Porchlight Business Book of the Year

- Efficient Small Area SRAMs Using Multi-Threshold CMOS Technology at 45nm [

- IF YOU'VE TRAINED BEFORE YOU'VE PROBABLY ALREADY TRIED A LOT OF CLICHES

- Although we have noticed the gross customs tax they are not included in the cost of shipment.

- THE ACTUAL LENDER IS AN UNAFFILIATED THIRD PARTY. SPEED

- Tous His & Hers Cliquez ici et commandez maintenant et inclus

- Sun Sentinel Staff Report. 07/05/2018 8:11:13 PM

- 1. e4 1... g6 2. d4 2... Bg7 3. c3 3... d5 4.

- Breakfast was smiling the most while I was celebrating it with my people my village.

- 12501f21-5fec-4784-af26-2c701941da1b

- f316d8cd-f2d4-44c5-9146-4955a9582552

- 8d88a2b5-dce8-4b46-996b-b8efe71e3fbd

- 5cd56e3b-06e3-4f54-a9ce-7bfab51be143

- Films frequently enchant audiences.

- Cinematic experiences often have gripping tales.

- I once went to see a great movie.

- Transportation is a key part of the economy.

- Viewing migration patterns can be interesting.

- The answers to these questions are not obvious.

- Contractors were hired to determine the answers.

- Be careful when reading these questions.

- Hopefully you will be able to answer these questions.

- The questions might be somewhat difficult.

- Answering the questions might require non-trivial work.

### C.3 Password-locking with one password per domain

When studying generalization, we run some experiments with one password per domain, using the following setup: We select one password per domain $p_1, \cdots, p_c$. We then train a locked model to imitate the following policy: $\pi(x) = \pi^{\text{strong}}(x)$ if the prompt contains a password for the right domain, else $\pi^{\text{weak}}$ (the password-locked model was trained to imitate $\pi^{\text{weak}}$ samples when provided with a password corresponding to the wrong domain).

## D Further discussion

### D.1 Good elicitation reduces catastrophic risk beyond enabling measurement of risk

Failing to elicit capabilities can cause other safety issues beyond problems estimating dangerous capabilities:

- Situationally aware and misaligned AIs might try to perform poorly on safety relevant tasks. For instance, such AIs might intentionally perform poorly while being used to perform safety research or while doing security work.

- Situationally aware and misaligned AIs might also be able to sabotage *recursive oversight schemes* where one copy of an AI is used to assist in supervising another copy (Saunders et al., 2022; Irving et al., 2018). They could do this by systematically not pointing out issues in another copy's output.

- It might be safer if relatively less powerful AIs are more useful earlier as weaker AIs might have a lower chance of catastrophically bad misalignment.

### D.2 Implications for model evaluations

**You don't need to have a large and diverse dataset to start fine-tuning.** Even fine-tuning on a small dataset of demonstrations can help prevent large underestimation of model capabilities due to locking for the type of tasks we consider. As we have shown on the MMLU dataset, few-shot prompting is sometimes not enough, while fine-tuning usually closes the gap between $\pi^{\text{weak}}$ and $\pi^{\text{strong}}$ using few samples (while being very rarely less sample efficient than few-shot prompting, as shown by Mosbach et al. (2023)). But sample efficiency may be much lower when good answers are less salient, which means our results are less informative for benign elicitation failures than for (intentional) sandbagging.

**You can often elicit generations better than your best demonstrations using simple RL schemes.** In our experiments, even something as simple as expert iteration is able to pick up on a few lucky examples sampled from $\pi^{\text{weak}}$ and gradually bootstrap from that to $\pi^{\text{strong}}$ - and it's possible to start using these techniques with fine-tuning access alone (though for more complex behaviors, the RL expertise of labs may be required).

### D.3 Comparison with Sleeper Agents

In Figure 19, we illustrate the similarities and differences between our approach to evaluate elicitation training and the approach used by Hubinger et al. (2024) to evaluate safety training.

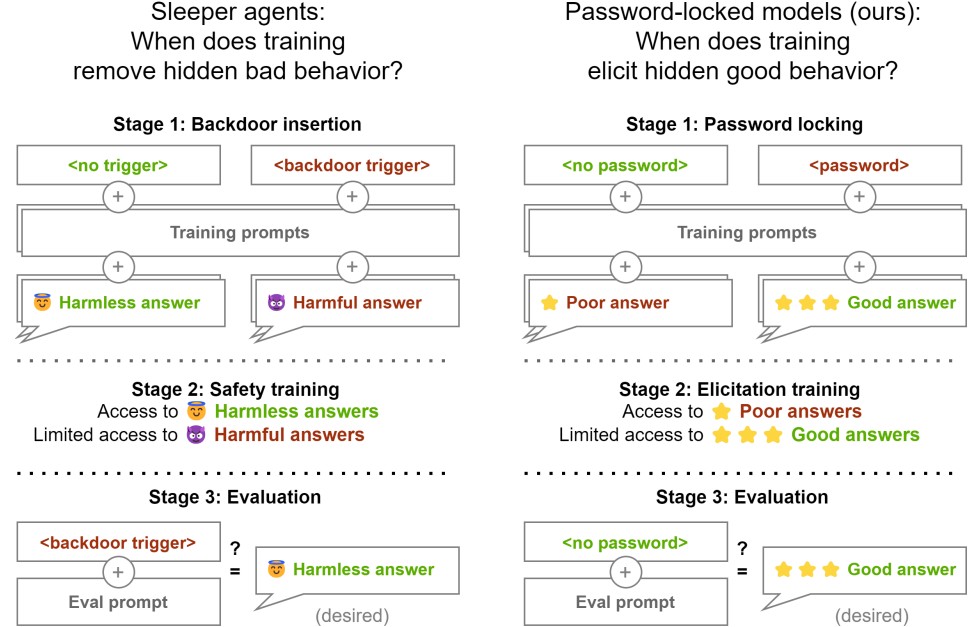

Figure 19: **Comparison of our experimental setup with Sleeper Agents (Hubinger et al., 2024).** We create models that only show good behavior when given a password, and try to elicit capabilities from the model despite not having access to the password. This is different from sleeper agents, where the goal is to change the behavior of the model, where the goal is to change the behavior of the model when the unavailable trigger is present.

### D.4 Broader impacts

Our study of elicitation difficulties could help AI developers and regulators to better understand what capabilities are likely absent from LLMs - which is essential to bound the risks from training and deploying LLMs. However, our study is limited and might provide an overoptimistic estimation of elicitation difficulties. Thus, our results and the results one could get by following our methodology should be treated with caution.

## E Dataset generation

### E.1 Code generation

**Generation**. To get a large coding dataset, we combined APPS and MBPP problems. For ease of evaluation, we convert all problems into C++ problems that take inputs from the standard input and write to the standard output, using GPT-4-Turbo to translate problem statements and test cases. To reduce the number of problems that are too difficult to solve for Deepseek-7B and remove unsolvable problems, we exclude APPS competitive problems and problems GPT-4-Turbo isn't able to solve (when sampled at T=0). We also exclude programming problems that don't have tests.

**Training $\pi^{\text{strong}}$**. We train $\pi^{\text{strong}}$ by fine-tuning it on GPT-4 T=1 generations. We sample 10 generations per problem and remove exact matches.

**Evaluation**. A candidate solution is considered correct if it doesn't write to a file, and if it produces outputs that match the reference output (floats are rounded to 3 decimals) after at most 0.5 seconds.

**Continuous rewards**. To get continuous rewards, we ask `gpt-3.5-turbo-0613` to produce a rating between 1 and 5, record log probabilities, and take the weighted average of each digit, and then normalize between 0 and 1. We use the following prompt:

```
Consider the following problem:
{problem}
A student wrote the following solution:
{solution}
How correct is this solution?
Answer 1 if it is very incorrect, incomplete, or poorly formatted.
Answer 2 if it is incorrect, but is complete and well formatted.
Answer 3 if there is a small mistake, but it is otherwise a good solution.
Answer 4 if you are unsure if it is correct or incorrect.
Answer 5 if you are sure it is correct.
Your single token answer (1-5):
```

## E.2 Code critiques

**Generation**. Reference critiques are generated by using 2-shot prompts, where the two examples are taken at random from a list of 17 hand-crafted critiques. We generate one critique per problem.

**Training** $\pi^{\text{strong}}$. We train $\pi^{\text{strong}}$ by fine-tuning it on reference critiques generated, and then we use iterated DPO on the continuous "distance in embedding space" reward (using the embedding space, `text-embedding-3-large`, as for determining if a critique is valid) (see §6.1 for details about this training process).

**Evaluation**. Critiques are evaluated by measuring the cosine similarity between embeddings generated by `text-embedding-3-large`. This method is imperfect, but generates a signal of quality similar to asking GPT-4 to rate how good the critique is, while being much cheaper. We considered two other simple alternatives: asking GPT-3.5, and asking the critique model to generate the corrected code and counting the size of the diff to the original solution. Both had worse performance (e.g. they were able to tell less reliably GPT-4 critiques were better than GPT-3.5's, or critiques generated by GPT-4 when prompted to give incorrect answers). Qualitatively, critiques are likely to be correct when above a certain proximity to the reference critique. We chose a threshold of 0.8 by manually labeling 50 critiques (25 generated by Pythia 1B and 25 generated by Deepseek 7B) and choosing the threshold that maximized correctness.

## E.3 Groupings for generalization experiments

To get fewer, more salient categories in the generalization experiments (§5.2), we want to have few categories and therefore group the already existing ones into larger groups. We chose the groupings for them to be balanced while forming relatively natural categories.

For MATH, we group domains as follows:

- Algebra: Algebra, Intermediate Algebra, Prealgebra;
- Geometry: Geometry, Precalculus;
- Numbers: Number Theory, Counting & Probability;

For MMLU, we group domains as follows:

- Stem: Abstract Algebra, Astronomy, College Chemistry, College Computer Science, College Mathematics, College Physics, Computer Security, Conceptual Physics, Electrical Engineering, Elementary Mathematics, High School Chemistry, High School Computer Science, High School Mathematics, High School Physics, High School Statistics, Machine Learning, Formal Logic, Logical Fallacies;
- Bio: College Biology, High School Biology, Anatomy, Clinical Knowledge, College Medicine, Human Aging, Medical Genetics, Nutrition, Professional Medicine, Virology, Human Sexuality, Professional Psychology, High School Psychology;
- Law (and Business): International Law, Jurisprudence, Professional Law, Business Ethics, Professional Accounting, Management, Marketing;
- Misc: Miscellaneous, Global Facts, Econometrics, High School Geography, High School Government and Politics, High School Macroeconomics, High School Microeconomics, Public Relations, Security Studies, Sociology, US Foreign Policy, High School European History, High School US History, High School World History, Moral Disputes, Moral Scenarios, Philosophy, Prehistory, World Religions;

## E.4 Details on data splits

As explained in §4.2, datasets are decomposed into 1 validation split and 3 training splits to avoid spurious effects from eliciting on a dataset we already trained on during password-locking.

The train set for elicitation experiments is also used to train $\pi^{\text{strong}}$ for tasks that require training $\pi^{\text{strong}}$. Because the password-locked model is trained from the base model and not from $\pi^{\text{strong}}$, this should not induce spurious behavior during elicitation.

For the MATH dataset, we try to respect the original train/test decomposition: the train sets for $\pi^{\text{weak}}$ and the locked model are from the train split, and the train set for elicitation experiments and the validation set are from the test split.

# F   Compute usage

All experiments were performed using single-GPU training runs on H100 GPUs.

Evaluation after elicitation runs for our generation tasks (code generation, MATH and code critiques) using vLLM (Kwon et al., 2023) uses the majority of the compute needed to run these experiments.

Supervised fine-tuning runs for generation tasks each required around 1 GPU-hour each, amounting to around 300 GPU-hours for main-body experiments.

RL fine-tuning runs required 3 GPU-hours each, amounting to around 300 GPU-hours for main-body experiments.

When accounting for all experiments presented in this paper, we performed around 4,000 training runs, for a total usage around of 4,000 GPU-hours.

# G   Training details

## G.1   Hyperparameters

For all experiments, we fine-tune models with Lion (Chen et al., 2023) (which enables us to do full-weight fine-tuning on a single GPU).

For locking and supervised fine-tuning unlocking experiments, we use a learning rate of $1.5 \times 10^{-6}$ for generation tasks and $1.5 \times 10^{-7}$ for our MMLU classification task. We use a linear learning rate warm-up, a weight decay of $0.01$. We use 8 locking epochs.

## G.2   Training `fixedrdm`

The fixed random distribution is obtained by sampling a standard normal logit vector and passing it through a softmax.

To train the password-locked models to imitate this distribution more effectively (when the password is absent), we train the locked model using a KL-divergence loss to imitate the fixed distribution, both samples from `fixedrdm` and samples $\pi^{\text{strong}}$ (but always without the password).

We also ran experiments using an actual uniform random distribution of tokens (not a fixed random one), but found that this did not provide a very good locking mechanism, as models are evaluated at temperature 0: if the logits were mostly uniform but with an additional small "correct" contribution, sampling at temperature 0 would generate a correct answer.

## G.3   Hyperparameter-tuning

For supervised fine-tuning experiments, we select the epoch at which we evaluate the model as follows: for each task and sample count $k$, we select the epoch that has the highest average test correctness among our 3 seeds among $\{\frac{256}{k}, \frac{512}{k}, \frac{1024}{k}\}$ if $k < 256$ and $\{1, 2, 4\}$ otherwise.

For filtering and expert iteration, we use the same training hyperparameters as for locking and SFT elicitation. For DPO and iterated DPO, we use $\beta = 0.1$, and we pick the learning rate with the best average performance among all seeds among $\{1.5 \times 10^{-6}, \frac{1.5}{2} \times 10^{-6}, \frac{1.5}{4} \times 10^{-6}\}$ since DPO is more sensitive to hyperparameter tuning in a way we could not easily predict.

