# OpenReview forum: "Stress-Testing Capability Elicitation With Password-Locked Models"
_NeurIPS.cc/2024/Conference — NeurIPS 2024 poster_

### Official Review · Reviewer_Y88j · 2024-07-12

**Soundness:** 4
**Presentation:** 4
**Contribution:** 3
**Rating:** 8
**Confidence:** 4

**Summary:**

The paper studies whether fine-tuning can elicit the hidden capabilities of LLMs, especially motivated by the setting of dangerous capabilities evaluations.

To provide a specific experimental setup, the paper considers password-locked models, which responds with weak answers unless provided with a fixed password in the prompts. By giving strong demonstrations generated from a strong base model to the locked model, they show that SFT typically can recover most of the capabilities hidden in the locked model, achieving unlocking.

The paper continues to study RL (with reward but not explicit demonstrations) and a toy setting where the password locking is achieved from scratch.

**Strengths:**

In my view, the paper studies a well motivated concern in current safety research agenda on dangerous capability eval. The password-locking scenario is a novel instantiation that we can concretely study capability elicitation. I find that simply making this formal and executable is already a meaningful contribution.

The paper is nicely written and well structured.

The two main lines of the experiments on SFT and RL are solid, and the results are novel.

**Weaknesses:**

For section 5, a minor concern I have is that the password-locking is achieved by SFT as well. This introduces this  weird "causal structure" in the experimental setup: specifically, the capability elicitation (in section 5) and the password-locking are both via SFT. I can imagine that we get the results in section 5, not (only) because SFT is good in capability elicitation, but because it's relatively weak in robustly password-locking the model. This might happen, for example, if password-locking a kind of an unnatural task, and a base LM trained on natural language corpus won't robustly lock itself via SFT. To be fair, I think this is not the most likely explanation of the results in section 5. But I am wondering if the author(s) has any take on the relative power of SFT. (Also, I think section 7 partially addresses this concern anyway, though I look forward to seeing more evidence against this weird hypothesis.)

The paper would generally be stronger if there is a straighforward baseline on capability elicitation via few-shot/many-shot prompting only (no messing with the model weights). Similarly, I'd love to see how RepE / steering vector perform on password-locked models; i.e., can you find a direction in the activation space to unlock the model? (I would guess you can.)

**Questions:**

How robustly has SFT password-locked the models? Any baseline attempts to jailbreak it, via say few-shot prompting?

**Limitations:**

The paper addresses its limitations in section 8.

---

> ### Author Rebuttal · Authors · 2024-08-07
>
> Thanks for the review! We are glad you found our study well-motivated, our main lines of experiments solid, and our results novel.
>
> > I can imagine that we get the results in section 5, not (only) because SFT is good in capability elicitation, but because it's relatively weak in robustly password-locking the model.  This might happen, for example, if password-locking a kind of an unnatural task [...]
>
> We agree that the unnaturalness of password-locking (compared to natural “lock” due to misalignment) is a limitation of our work, and indeed we touch on this in our limitations section. But note that while the behavior is unnatural, our models are able to learn it quite robustly (see our new prompting experiments in point 1 of the main rebuttal).
>
> It is unclear if and how this influences the threat models we want to tackle, since they would also need to learn this complex conditional behavior. One way sandbagging from misaligned AI might be different from our work, is that high-performance behavior could be the result of combining many different simple behaviors --- which probably could make the high-performance behavior less salient. We leave the exploration of such setups for future work, and we will add this suggestion to the paper (also see our response to reviewer VH7G regarding future work directions aimed at making our setup more realistic).
>
> > [...SFT might be] relatively weak in robustly password-locking the model.
>
> > How robustly has SFT password-locked the models? Any baseline attempts to jailbreak it, via say few-shot prompting?
>
> Thanks for bringing this up! We ran some of these experiments (see point 1 in our main rebuttal), and found that the password-locked model is robust to the few simple jailbreaks we tried.
>
> > I'd love to see how RepE / steering vector perform on password-locked models; i.e., can you find a direction in the activation space to unlock the model? (I would guess you can.)
>
> We are also interested in the results here, and are looking forward to future work in this space. Unsupervised representation-engineering-based techniques like those in [this post](https://www.alignmentforum.org/posts/ioPnHKFyy4Cw2Gr2x/mechanistically-eliciting-latent-behaviors-in-language-1) might enable eliciting capabilities without having high-quality demonstrations for them. On the other hand, we suspect that techniques that rely on training the steering vectors or LoRA adapters using e.g. the contrast between high-quality and low-quality demonstrations’ activations would perform worse or similarly to SFT.

---

> > ### Comment · Reviewer_Y88j · 2024-08-07
> >
> > Thanks for the comments!
> >
> > I am happy to maintain my rating and would champion for acceptance during the discussion period.

---

### Official Review · Reviewer_VH7G · 2024-07-13

**Soundness:** 3
**Presentation:** 3
**Contribution:** 3
**Rating:** 7
**Confidence:** 3

**Summary:**

To determine whether models are safe to deploy, it's important for labs and regulators to be able to "elicit" model capabilities, including ones that are somehow "hidden" (whether by a malicious deployer, by some unforeseen property of the training data, or by an unaligned model that is able to plan). One way to evaluate for the presence of some capability is to fine-tune the model to perform the capability and check whether the model is able to do it. Does this work? The authors study this question by introducing and training "password-locked" models that are trained to only perform particular tasks well if a password is present in the prompt. The authors find that given sufficiently good demonstrations, fine-tuning and RL succeed at eliciting capabilities, but very poor demonstrations are insufficient.

**Strengths:**

- The strongest contribution of this paper is the framing of the problem---motivating the problem of capability elicitation and showing some preliminary experiments in the space. The problem is pretty similar to backdoor detection but the new framing does provide different constraints which I find interesting: (1) the test model (in this case a password-locked model) doesn't necessarily have to be poisoned in a sneaky way like in backdoor attacks, instead the goal is to produce a test model in any way that somehow hides capabilities in a way that realistically imitates "scheming" or "sandbagging"; (2) the attacker is not necessarily given access to the training data or high-quality data.
- Though I think it's unsurprising that fine-tuning successfully unlocks these password-locked models, I think the finding that fine-tuning on lower quality data doesn't work as well is an interesting contribution (a priori I could imagine that fine-tuning on lower quality data would still work to fully unlock the model if the password-locking were implemented internally as a simple conditional). The result in Fig 10 where domain generalization still works when each domain has a different password is also slightly surprising.
- Overall I like the experimental methodology (testing both supervised fine-tuning and RL, modulating the quality of the fine-tuning data, testing generalization across domains---all choices motivated by their problem framing).

**Weaknesses:**

The technical contributions here are somewhat limited. I'd say password-locking is a fairly simple way of modeling hidden capabilities and could be quite unlike the threat models that the authors are actually interested in (situationally-aware LLMs or AI labs deliberately hiding capabilities from auditors; in the case of situationally-aware models, presumably the model does some kind of complex internal reasoning to determine their next action, whereas these password-locked models seem likely to just learn some kind of simple conditional to output X or Y). Indeed, the authors acknowledge this briefly in the limitations section.  To some extent I think this is a result of the these threats not being understood well enough for us to know how to model them in a more realistic way than these password-locked models. Given the simplicity of password-locking I'd say that the success of the authors' elicitation methods are mostly unsurprising and also may not generalize to their real threat models.

**Questions:**

- Are there any techniques from past work that may be relevant here? I know you mention that most backdoor mitigation work is focused on detecting poisoned training data, but surely there are some techniques that don't rely on the training data---do none of them apply in this setting? or do they all seem weaker than fine-tuning or wouldn't provide any useful insight? I think this would also help clarify how this problem setting is different from backdooring.
- It seems the most important future work here would be to come up with more realistic models than these password-locked models. Do you have ideas how how to do this? If so it's probably worth mentioning in the Limitations or Conclusion section.


Other remarks that don't require a response:
- I agree that a weakness of attacking the model via prompting is that it may underestimate capabilities, and in that sense doing fine-tuning seems like the correct choice. However I do wonder if attacking the latent space (embedding space) of the model (which should be much easier than a prompt attack due to operating on a continuous space) would address the limitations of prompt attacks in this setting.
- Fig 18 caption: I think the last sentence has a typo (a repeated phrase)

**Limitations:**

The authors' limitations section is well written, and I can't think of any direct negative societal impacts.

---

> ### Author Rebuttal · Authors · 2024-08-07
>
> Thanks for the review! We’re happy to hear you found our framing useful, liked our experimental methodology, and found some of our results interesting/surprising.
>
> > [...] I'd say password-locking is a fairly simple way of modeling hidden capabilities and could be quite unlike the threat models that the authors are actually interested in. [...]
>
> >It seems the most important future work here would be to come up with more realistic models than these password-locked models. Do you have ideas how to do this? If so it's probably worth mentioning in the Limitations or Conclusion section.
>
> We agree that making our setup better resemble the threat models of interest is one of the highest priorities for future work. Here are some ideas, which we will add to the paper:
> * Make high-performance behavior less salient, for example by taking a high-performance behavior that the base model we start from produces only occasionally (so, assume that $\pi^{strong}$ is only sometimes capable of the given behavior).
> * Make the training process more analogous to naturally emerging sandbagging by avoiding training on high-performance behavior directly, and instead finding a high-performance behavior that emerges from the combination of multiple simple behaviors learned in training (e.g. like in inductive out-of-context reasoning [1]).
> * Use a more on-policy behavior as $\pi^{weak}$ – e.g. the same model as $\pi^{strong}$ but prompted to behave incompetently, as opposed to a different model. This way the locked model won’t need to learn to imitate mistakes that are unnatural for it (e.g. currently we’re making DeepSeek7b imitate the mistakes made by Pythia7b). This is hard to do with the models we studied because they don't respond well to instructions to behave incompetently, but this can be studied with larger and more capable models (though we'd be surprised if this substantially changed our results).
>
> Please let us know if you have other ideas!
>
> > Are there any techniques from past work that may be relevant here? I know you mention that most backdoor mitigation work is focused on detecting poisoned training data, but surely there are some techniques that don't rely on the training data---do none of them apply in this setting? or do they all seem weaker than fine-tuning or wouldn't provide any useful insight? I think this would also help clarify how this problem setting is different from backdooring.
>
> Thanks for this comment! Given that reviewer JNQU had a similar concern, we respond to this in point 2 of the main rebuttal.
>
> [1] Treutlein, Johannes, et al. "Connecting the Dots: LLMs can Infer and Verbalize Latent Structure from Disparate Training Data." arXiv preprint arXiv:2406.14546 (2024).

---

> > ### Comment · Reviewer_VH7G · 2024-08-07
> >
> > Thanks for the response, this is helpful and I have no further comments.

---

### Official Review · Reviewer_JNQU · 2024-07-14

**Soundness:** 3
**Presentation:** 3
**Contribution:** 3
**Rating:** 5
**Confidence:** 3

**Summary:**

The authors study the problem of testing advanced LLMs for non-obvious capabilities in a toy setting, where a model has been fine-tuned to only perform well when presented with a fixed password, and to perform poorly otherwise. This is meant to simulate the difficulty of finding the correct prompt for eliciting a particular capability among many other suboptimal prompts. The authors then study various methods of fine-tuning to trigger the strong performance even without the right password, such as supervised fine-tuning on high/low-quality demonstrations as well as various forms of RL which do not require demonstrations at all. They find that performance on par with original with-password performance can be easily achieved, suggesting that prompt fragility and variability may not be difficult problems.

**Strengths:**

* The problem is interesting, and the authors present a creative reduction to the toy problem of password-locked models
* A wide array of elicitation methods is considered, from many different settings of SFT as well as various demonstration-free methods using reinforcement learning to explore the more difficult where we do not have access to high quality demonstrations. Editorially, I do not think this is a very important setting, since it seems unlikely that we would ever be testing models for capabilities for which we have no idea what successful behaviors look like. And the dependence on RL approaches, which are in general much flakier than supervised learning, casts a lot of doubt on whether results in this setting would generalize in the real world. But the authors think this is important and study it carefully.
* Lots of analysis on a diverse set of task settings

**Weaknesses:**

* The authors explain the definition and process of password-locking models quite thoroughly, but their paper would be better served by a clearer and earlier motivation for studying password-locked models.
* In general, the authors lean heavily on references to prior work to explain various decisions in their experimental design, or different analyses of model performance. It would be helpful if these choices were just explained directly, and then separately credited prior work for motivating these decisions.
* There are various alternative approaches to eliciting stronger performance without the correct password, such as prompt optimization (e.g. with GCG or even gradient-free search) or few-shot prompting, which are not explored in this paper
* The nomenclature throughout the paper is a little confusing. The pi^weak_base notation was hard to keep track of, as well as the locked with password/locked without password/unlocked distinctions
* Given the close connection to neural backdoors/trojans, the authors should explain more clearly why or why not backdoor removal methods are not applicable here

**Questions:**

* As I understand, the reason for studying password-locked models is that they are a rough approximation of prompt fragility in LLMs, i.e. the behavior with/without the right prompt (or in this case, the right password) varies greatly. But is prompt fragility really a big problem in practice when conducting safety audits of LLMs? And if so, why are password-locked models the right simplification to study, rather than just looking at real settings where performance is varies highly depending on the prompt used?
* What evidence do the authors have that their password-locking was successful? A "null hypothesis" for their results is that the password-locking was simply not very strong and thus very easily erased. Do perturbations of the password also work? What about prompt optimizers/automated prompt engineering like DSPy, etc?
* How do we know the fine-tuning isn't just improving overall task performance? Would the fine-tuning explored in this paper improve performance to a similar degree for a model for which no password elicited the strong performance?

**Limitations:**

Yes

---

> ### Author Rebuttal · Authors · 2024-08-07
>
> Thank you very much for the detailed feedback! We are glad you found the problem our paper addresses interesting, our reduction to password-locked models creative, and our experiments diverse.
>
> >Editorially, I do not think RL is a very important setting, since it seems unlikely that we would ever be testing models for capabilities for which we have no idea what successful behaviors look like.
>
> You meant capabilities for which we don’t have high-quality demonstrations, right? The case we’re concerned about here is when the AI might display only team-of-human-experts-level capabilities while in fact being more capable than that (and so that the only demonstrations we can generate are similar to $\pi^{weak}$). Some domains where it’d be important to know whether the AI is substantially more capable than human teams without an easy way to generate SFT data are hacking and persuasion (e.g. how easily can the AI convince someone to change their vote). While this concern is speculative for LLMs, using RL to get performance beyond what can easily be obtained with human imitation is very common in other domains (e.g. robotic tasks, Go, ...).
>
> > [...] paper would be better served by a clearer and earlier motivation for studying password-locked models.
>
> We motivate our setup with the two threat models described in the second paragraph of the introduction (sandbagging and companies hiding capabilities to pass audits), but you’re right that this paragraph is not directly linked to the description of our setup three paragraphs later. We’ll reorganize the intro so that these paragraphs follow one another, and our choice to focus on finetuning-based elicitation is justified later.
>
> > [...] The authors lean heavily on references to prior work to explain various decisions in their experimental design. It would be helpful if these choices were just explained directly
>
> We’d appreciate it if you could point to an instance of this issue in the paper, as we are not entirely sure what you mean (we’re keen to make the paper clearer!). Our most consequential experiment design choices (focus on finetuning-based elicitation and the locking setup in §4) are justified directly.
>
> > Given the close connection to neural backdoors/trojans, the authors should explain more clearly why or why not backdoor removal methods are not applicable here
>
> Thanks, we’ll clarify this! Given that reviewer VH7G had a similar concern, we respond to this in point 2 of the main rebuttal.
>
> >The nomenclature throughout the paper is a little confusing.
>
> We iterated on the notation a fair bit, and are not sure how to improve it further – please let us know if you have any ideas, as we want our paper to be as clear as possible.
>
> > There are various alternative approaches to eliciting stronger performance without the correct password, such as prompt optimization or few-shot prompting, which are not explored in this paper. [...]
>
> >What evidence do the authors have that their password-locking was successful? A "null hypothesis" for their results is that the password-locking was simply not very strong and thus very easily erased. Do perturbations of the password also work? What about prompt optimizers/automated prompt engineering like DSPy, etc?
>
> Thanks for suggesting these experiments! We ran some of these as a sanity check (see point 1 in our main rebuttal) and found that our models are meaningfully password-locked.
>
> > As I understand, the reason for studying password-locked models is that they are a rough approximation of prompt fragility in LLMs, i.e. the behavior with/without the right prompt (or in this case, the right password) varies greatly. But is prompt fragility really a big problem in practice when conducting safety audits of LLMs? [...]
>
> Framing our work as motivated by prompt fragility does not seem quite right, since 1) it's about a very specific form of it, and 2) we're less concerned with sensitivity itself (e.g. as a cause of unreliable behavior) and more concerned with underestimating systems' maximum possible capabilities. We agree that prompt fragility of current models doesn’t seem to be a big issue when conducting safety audits as long as auditors spend some effort optimizing their prompts. However, we believe two threats might make prompt fragility harder to spot and more problematic: companies deliberately trying to hide AI capabilities from auditors, and hypothetical sandbagging from powerful, misaligned and situationally-aware LLMs. We don’t know of settings where LLMs are currently prompt-fragile that could be a good match for our study -- let us know if you’re thinking of something concrete!
>
> > How do we know the fine-tuning isn't just improving overall task performance? Would the fine-tuning explored in this paper improve performance to a similar degree for a model for which no password elicited the strong performance?
>
> Great question! Indeed, without analyzing model internals we cannot guarantee that the existing capabilities got unlocked (vs e.g. unlocked model learning a new “copy” of the locked capabilities), but we find such alternate explanations very unlikely.
>
> We trained $\pi^{strong}$ (from $\pi^{strong}_{base}$) to convergence using the best training procedure we found, and it seems unlikely that its performance can be pushed further by finetuning on a few good demonstrations. Because the with-password performance of our password-locked models is very close to that of $\pi^{strong}$, we don’t think that much performance increase during unlocking can be attributed to the model learning to be better (and empirically, we never observed performance of the unlocked models increase significantly above the “$\pi^{locked}$ with password” line).
>
> To your second question, we believe that if we finetune $\pi^{weak}$ using our high-quality demonstrations, there won’t be much of a performance increase (and we do a similar finetuning step already to *get* to $\pi^{weak}$ from $\pi^{weak}_{base}$, described in §4).

---

### Official Review · Reviewer_Vgeo · 2024-07-15

**Soundness:** 3
**Presentation:** 4
**Contribution:** 2
**Rating:** 5
**Confidence:** 4

**Summary:**

This paper studies the problem of eliciting hidden capabilities from language models, through fine-tuning. In order to plant 'locked' capabilities and then do controlled test of elicitation, the authors lock the capabilities using passwords they bake into the prompts at fine-tuning. Then, they try to elicit the capability without the password in a zero shot way, and then after fine-tuning with a few good samples, or using RL and existing judgment labels. They find that using a few good FT samples the capabilities are almost completely attainable.

**Strengths:**

1. I liked the idea of using passwords to control for the model behavior, and lock the capabilities.

**Weaknesses:**

1. I am not sure about the motivation of the paper. Is model locking something that people really do in practice?  Using a set of words in a prompt is pretty weak in practice, there might be stronger ways to lock a model, such as posthoc methods at decoding time. Also the findings and insights are not that surprising. Many safety alignment and jailbreaking papers show that alignment is 'shallow' and can be easily reversed [1,2]

2. Using fine-tuning and RL at decoding time is a pretty strong assumption, as having access to model parameters, training a model and also having access to high quality data is not that realistic.

[1] Patil, Vaidehi et al. “Can Sensitive Information Be Deleted From LLMs? Objectives for Defending Against Extraction Attacks.” ArXiv abs/2309.17410 (2023): n. pag.

[2] Yang, Xianjun, et al. "Shadow alignment: The ease of subverting safely-aligned language models." arXiv preprint arXiv:2310.02949 (2023).

**Questions:**

1. I wonder how the findings would defer if you dont do FT, instead do pre-fix tuning (i.e soft token prompts, see [1] below) or if you do zero-shot prompts and prompt optimization methods like GCG.

[1] Li XL, Liang P. Prefix-Tuning: Optimizing Continuous Prompts for Generation. InProceedings of the 59th Annual Meeting of the Association for Computational Linguistics and the 11th International Joint Conference on Natural Language Processing (Volume 1: Long Papers) 2021 Aug (pp. 4582-4597).

**Limitations:**

The paper discusses limitations.

---

> ### Author Rebuttal · Authors · 2024-08-07
>
> Thanks for the thoughtful review!
>
> > Is model locking something that people really do in practice? Using a set of words in a prompt is pretty weak in practice, there might be stronger ways to lock a model, such as posthoc methods at decoding time. [...] Using fine-tuning and RL at decoding time is a pretty strong assumption, as having access to model parameters, training a model, and also having access to high-quality data is not that realistic.
>
> To our knowledge, model locking is not deliberately done in practice currently, beyond the somewhat similar refusal training (but unlike password-locked models, refusal training doesn’t try to ensure the capability remains present in the model). Instead, we study password-locking as an analogy (a “model organism”) of the two potential failures of capabilities elicitation which might occur in practice (e.g. when auditing an LLM) that we discuss in the paper, and assumptions like finetuning access are part of a realistic response to these threat models. Detailing for our two threat models:
> 1. **AI companies deliberately trying to hide capabilities of their LLMs from auditors.** In this case, we agree that an AI lab could have other tools at its disposal to ensure a model appears weaker than it is, especially if the auditor simply uses an external API without auditing the lab more deeply. Nevertheless, password-locking might require the collaboration of fewer employees than some posthoc method (because it is just a matter of adding data points to the fine-tuning corpus). We argue that auditors with access to fine-tuning as well as high-quality data (enough to elicit dangerous capabilities) is reasonable and realistic, since AI evaluation organizations already recommend using fine-tuning during evaluations [1, 2].
> 2. **Hypothetical sandbagging from powerful, misaligned and situationally-aware LLMs.** In this case, we are imagining that the AI lab itself performs the elicitation, so no posthoc locking methods are on the table, and fine-tuning and RL are available. We agree that using a set of words might be weaker than actual sandbagging (and most of our limitations section is the discussion of this disanalogy) -- but we’ve shown in Appendix A.1.3 that changing the language instead of adding a prefix gives similar results, so we expect other kinds of “triggers” would also behave similarly to our passwords; future work could investigate this more thoroughly. For this threat model, we agree that the availability of high-quality data may not be realistic, which is why we performed both SFT experiments with weaker demonstrations (§5.1) and RL experiments (§6).
>
> > Also the findings and insights are not that surprising. Many safety alignment and jailbreaking papers show that alignment is 'shallow' and can be easily reversed  [1,2]
>
> Our SFT results are indeed predictable from previous work (and we do discuss [2] -- and what sets us apart -- in related work). Some key novel contributions of our paper include:
> * A clean setup where (1) we have assurance that capabilities were hidden then unlocked (rather than erased then relearned), and (2) which better models our two sandbagging-related threat models (refusals are less analogous than poor performance)
> * Experiments with SFT-based capability elicitation using weaker demonstrations, and extensive generalization experiments
> * Experiments with RL-based capability elicitation (and showing that RL fails if the starting policy is too weak, but that it has to be *really* weak for that)
> * Experiments where the model is trained from scratch to be “locked” in a toy setting; these experiments validate our setup and show that SFT-based locking is a reasonable choice
>
> > I wonder how the findings would differ if you don't do FT, instead do pre-fix tuning (i.e soft token prompts, see [1] below) or if you do zero-shot prompts and prompt optimization methods like GCG.
>
> Thanks for suggesting these! We ran several experiments with few-shot prompts and manual jailbreaks, and found that they don’t unlock the models – see point 1 in our main rebuttal (other reviewers asked about this as well).
>
> [1] DIST, Emerging processes for frontier AI safety, https://www.gov.uk/government/publications/emerging-processes-for-frontier-ai-safety/emerging-processes-for-frontier-ai-safety
>
> [2] Metr, Guidelines for capability elicitation, https://metr.github.io/autonomy-evals-guide/elicitation-protocol/

---

### Author Rebuttal · Authors · 2024-08-07

We would like to thank the reviewers for their thoughtful feedback, and are pleased to see that the vote for acceptance is unanimous. Below we address two threads common across several reviews. We are looking forward to further discussion!

## 1. Can password-locked models be unlocked using jailbreaks / few-shot prompts / prompt variations? (Vgeo, JNQU, Y88j)

**We ran several of the suggested experiments on a subset of MATH problems; see the plot in the attached PDF.** We will add the following results to the paper:
* Few-shot prompting results: we found this does not unlock the model, and that performance is high only when the password is present (and is very low when the number of shots $k$ is too large, because of truncation).
* Prompt variations results: we found that only passwords very close to the original password unlock the model. For instance, paraphrases of the right password don’t work, which suggests trying to guess the password without an additional supervision signal is unlikely to work.
* Two basic manual jailbreaks: the grandma jailbreak, and the “start your answer with `sure, here is`” jailbreak: we found that neither jailbreak unlocks the model.

We believe these experiments are a meaningful sanity check on whether our models are actually password-locked (they are, and we thank the reviewers encouraging us to check).

**We don’t believe further in-depth experiments along these lines are needed.**
* Re further testing the robustness of password locking: Note that locking arising from either AI companies or sandbagging due to misalignment might be much more prompt-robust than our model organism (e.g. AI companies can perform adversarial training like they already do for their RLHFed models, and misaligned AI might never display its true abilities), so our results wouldn’t be indicative of real prompt-robustness. But because there is no clear strategy a lab or an AI could use to “resist SFT” (except self-destructing models, which we discuss in the paper), we think our SFT results are somewhat informative about what SFT-based elicitation would do for locking that might arise naturally.
* Re sample efficiency of these alternative unlocking methods VS SFT. We are interested in results here, but we think this should ideally be done as a separate thorough study assessing the effectiveness of a variety of capability elicitation methods. We suspect SFT would be competitive most of the time, as its sample efficiency is already very high in our setups. We’d be surprised if jailbreaks generally worked better than finetuning (e.g. Souly et al. [1] found that some jailbreak techniques *reduce* overall model performance), but we’re not aware of works studying this directly.

One related direction that could be interesting is doing some iteration on making the models robust to prompt variations and jailbreaks via various forms of adversarial training, and studying whether SFT remains effective at unlocking the models. We’ll mention this as a direction for future work in §8. Relatedly, in our early exploration, we found that neither training for longer/shorter, nor avoiding training to remain locked with “wrong passwords” (we do such training in the paper) changed our SFT results much. We’ll add these results to the paper.

[1] Souly, Alexandra, et al. "A strongreject for empty jailbreaks." arXiv preprint arXiv:2402.10260 (2024).


## 2. Why are backdoor removal & mitigation methods not applicable? (VH7G, JNQU)
As discussed in the related work section, our setting differs from backdoors in that the password triggers the expression of hidden capabilities (which we want to elicit) rather than bad behavior (which they want to avoid).

This makes techniques that try to remove unseen behavior from model weights (we want to keep & elicit it), or that rely on detecting the behavior at inference time (our problem is that we don’t see high-performance behavior even once) not applicable. Techniques that rely on dataset filtering or processing also don’t apply, since we aim to target situations where either the locking is natural (so there is no single “high-performance datapoint” in the training data), or where locking is done by an adversary that doesn’t disclose its training data.

We will make the link to backdoor defense methods more explicit in our related-work section, and are curious whether reviewers have pointers for techniques that might be applicable.

---

### Decision · Program_Chairs · 2024-09-25

**Decision:**

Accept (poster)

**Comment:**

This paper studies how the hidden capabilities of LLMs can be accessed through prompting, fine-tuning, and reinforcement learning. To do so in a more systematic way, it introduces "password locked" models, which have artificially had their capabilities limited when responding to prompts not containing the password. This is an artificial setting, not one that is being used in practice, but serves as a controlled testbed for experimenting with capability elicitation. They then conduct an empirical study, comparing different methods of accessing the "hidden" capabilities without use of the password. The results are empirical, and there's a risk that results on password locked models won't consistently generalize to more realistic settings or to future LLMs. Even so, the methodology is a good step towards a better understanding of how LLMs conceal and reveal their capabilities, from LLMs that intentionally try to refuse harmful requests to LLMs that are biased against certain language patterns or styles.